# Honey-collecting in prehistoric West Africa from 3500 years ago

Julie Dunne [1✉], Alexa Höhn [2], Gabriele Franke[2], Katharina Neumann [2✉], Peter Breunig[2], Toby Gillard[1], Caitlin Walton-Doyle[1] & Richard P. Evershed [1✉]

Honey and other bee products were likely a sought-after foodstuff for much of human history, with direct chemical evidence for beeswax identified in prehistoric ceramic vessels from Europe, the Near East and Mediterranean North Africa, from the 7th millennium BC. Historical and ethnographic literature from across Africa suggests bee products, honey and larvae, had considerable importance both as a food source and in the making of honey-based drinks. Here, to investigate this, we carry out lipid residue analysis of 458 prehistoric pottery vessels from the Nok culture, Nigeria, West Africa, an area where early farmers and foragers co-existed. We report complex lipid distributions, comprising *n*-alkanes, *n*-alkanoic acids and fatty acyl wax esters, which provide direct chemical evidence of bee product exploitation and processing, likely including honey-collecting, in over one third of lipid-yielding Nok ceramic vessels. These findings highlight the probable importance of honey collecting in an early farming context, around 3500 years ago, in West Africa.

[1] Organic Geochemistry Unit, School of Chemistry, University of Bristol, Bristol, UK. [2] Goethe University, Institute for Archaeological Sciences, Frankfurt am Main, Germany. ✉email: julie.dunne@bristol.ac.uk; k.neumann@em.uni-frankfurt.de; r.p.evershed@bristol.ac.uk

Honey, a rare source of sweetness, was likely a much sought-after foodstuff for much of human history. Recognition that bee products, including honey and larvae, offered a high-quality source of dietary energy, fat, and protein explains the long history of bee exploitation in the hominin lineage[1,2]. Honey is energetically dense[3,4] and easy to consume and digest and thus may have contributed to potential links between nutrition and neural expansion of the enlarging hominin brain[3]. Our closest living relatives, the chimpanzees (*Pan troglodytes*), forage for honey (and brood) when it is available, as do baboons and other great apes[5], suggesting the importance of honey extraction in the emergence of complex tool use in hominoids[6].

Today, honeybees are an integral part of socio-ecological landscapes and beekeeping plays an important global economic role with around 1.6 million tonnes of honey being produced annually[7]. Wild honey is also known to be widely collected by foragers globally, except in environments such as the Arctic and Subarctic where bees do not survive[2]. However, evidence for ancient human exploitation of the honeybee is rare, save in palaeolithic rock art depicting bees and honey, found in Spain, India, Australia and Southern Africa, spanning the period 40,000–8000 years ago. The majority of prehistoric rock art, with over 4000 sites portraying bees, honeycombs and honey-collecting, is located in Africa, at Didima Gorge in Namibia and other locations[8–10]. Furthermore, the archaeology of honey-collecting is largely invisible, in contrast to the, often excellent, survival of other organic materials such as animal bones or plant remains[11].

Recently, lipid residue analysis identified evidence for the presence of beeswax in prehistoric pottery vessels from across Neolithic Europe, the Near East and Mediterranean North Africa, providing evidence for human exploitation of the honeybee from at least the seventh millennium BC[12]. However, little is known of its importance in other areas globally, for example, in the subsistence of hunter-gatherer groups and early farming communities in West Africa, despite considerable historical and ethnographic literature on the use of honey across Africa, both as a food source and in the making of honey-based drinks, alcoholic or otherwise[2,13,14]. Today, honeybee hive products, including honey, propolis and pollen, for food and medicinal purposes, support livelihoods and provide sources of income for local communities across much of Africa, through both beekeeping and wild harvest[2,7,15]. In the West African tropical rain forest, collecting wild honey, found in natural hollows in tree trunks and on the underside of thick branches, is a common subsistence activity[15–17].

West Africa is a vast geographic area extending from the Atlantic Ocean in the west, and south to Cameroon, comprising a range of ecological zones encompassing expanses of coast, rainforest, woodlands and grasslands. The archaeological record in this region comprises a highly complex and diverse, but patchy, mosaic, meaning our understanding of processes such as the transition to food production are not fully understood[18–22]. The earliest known pottery in Africa originates from West Africa, at the site of Oun-jougou, Mali, in the 10th millennium cal BC, likely invented for the processing of wild grasses[23], with a much later adoption of domesticated animals and development of cultivation in the region. Indeed, the spread of food production in West Africa seems to have been uneven with farming and agro-pastoral communities coexisting with hunter-gatherer-fisher groups[18,20–22,24–26].

One of the most well-known cultures in prehistoric West Africa is the Central Nigerian Nok culture (Fig. 1), characterised by its remarkable terracotta figurines, which constitute the earliest large-sized figurative art objects in Africa outside Egypt[27] (Fig. 2) and early evidence for iron production in West Africa, ca. first millennium BC[28]. The Nok culture spans a period of around 1500 years, beginning around the middle of the second millennium

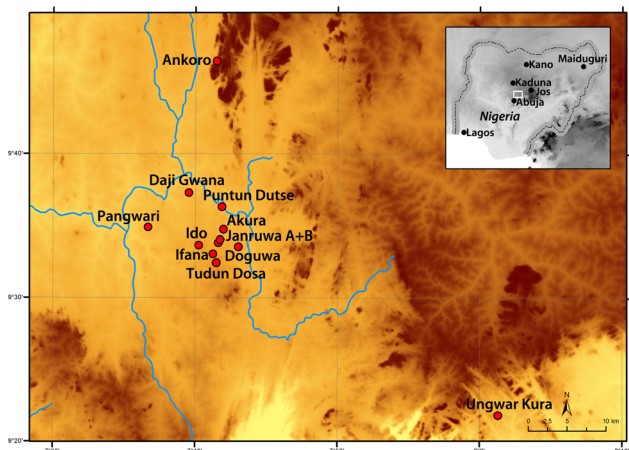

**Fig. 1 Map of Nok sites.** Distribution map showing Nok archaeological sites sampled in this study, including Akura, Ankoro, Daji Gwana, Doguwa, Ido, Ifana, Janruwa A, Janruwa B, Puntun Dutse, Pangwari, Tudun Dosa and Ungwar Kura (map by Eyub F. Eyub).

BC[29,30]. Inhabitants of Nok settlements appear to be agriculturalists existing on a diet that included millet (*Pennisetum glaucum*) and cowpea[31] (*Vigna unguiculata*). A complete absence of animal bones, due to acidic soils, means that it is not known whether the Nok people kept domesticated animals or depended solely on hunting wild game.

Here we perform organic residue analysis on 458 potsherds from 12 Nok archaeological sites (Fig. 1) to investigate Nok diet and subsistence. Lipid biomarker analyses by gas chromatography/mass spectrometry (GC/MS) yield a complex suite of lipids including *n*-alkanes, *n*-alkanoic acids and fatty acyl wax esters, indicating the presence of beeswax in over one-third of Nok lipid-yielding vessels. Hence, we report direct chemical evidence of beeswax, likely related to honey processing, in West African ceramic vessels, and thus honey-collecting, in an early farming context, around 3500 years ago, and, additionally, biomolecular evidence for the palaeoecological range of *Apis mellifera adansonii* in Holocene West Africa.

## Results

**Sampling.** A total of 458 potsherds from 12 sites in Central Nigeria, covering the Early, Middle and Late Nok periods (Fig. 3), were analysed using established protocols described in detail in earlier publications[32,33]. Vessels sampled comprised everted rim pots with body diameters of ca. 20–30 cm, the common form found within the Nok pottery assemblage. Recovery rates were low overall, with 66 sherds (14.4%) yielding interpretable lipid profiles (Table 1). Lipid concentrations ranged from 7 to 1815.6 μg g$^{-1}$, with one exception, sherd NOK284, displaying a typical degraded animal fat profile, which yielded a lipid concentration of 13.2 mg g$^{-1}$. Lipid biomarker analyses by GC/MS showed the Nok residues to fall into three broad categories. The first group of lipid profiles (n = 14) were dominated by the free fatty acids, palmitic (C$_{16:0}$) and stearic (C$_{18:0}$), typical of a degraded animal fat[34,35]. A further 27 lipid profiles (group 2) comprised a range of complex distributions denoting the processing of various plant types.

**Beeswax lipid profiles.** The remaining 25 lipid profiles (Table 2) comprised very distinctive series of even-numbered *n*-alkanoic acids (C$_{20}$–C$_{32}$), *n*-alkanols (C$_{22}$–C$_{34}$), and *n*-alkanes (C$_{23}$–C$_{35}$), indicative of the presence of beeswax. Lipid concentrations ranged from 14 to 1815.6 μg g$^{-1}$ and varied significantly with

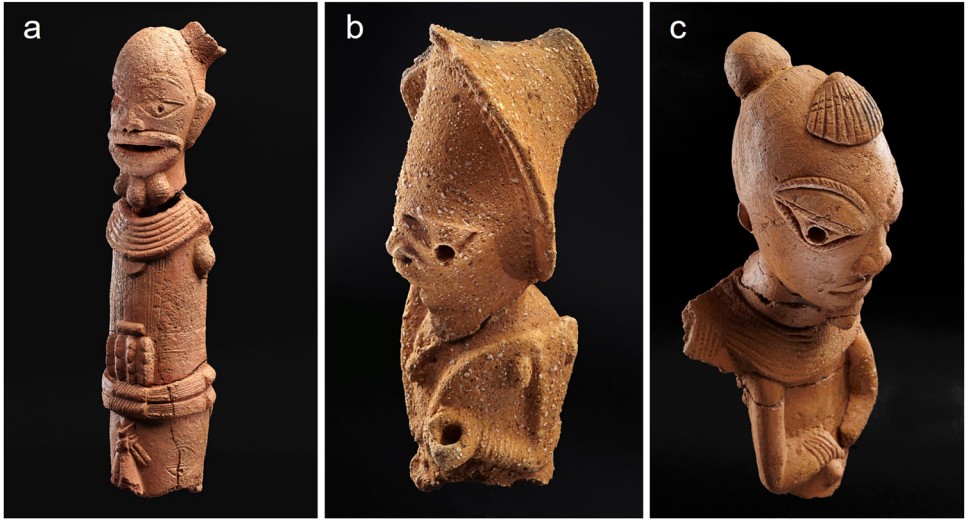

**Fig. 2 Nok culture terracotta statues.** Composite of Nok culture terracottas from three sites: **a** Ifana, **b** Ungwar Kura and **c** Pangwari (© Goethe University, Frankfurt).

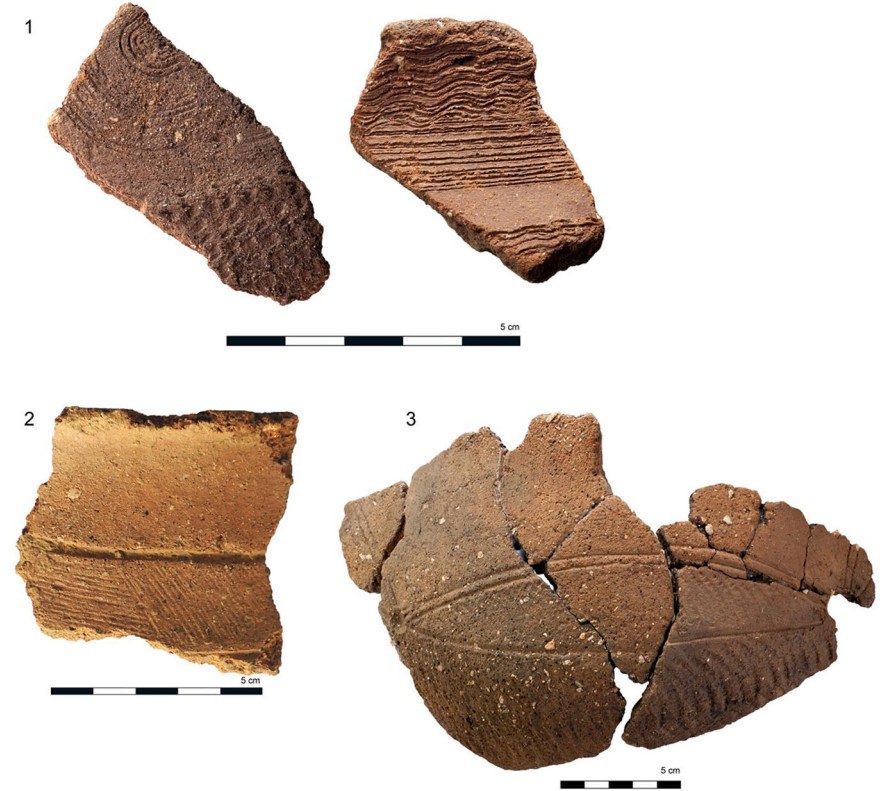

**Fig. 3 Typical Nok potsherds. 1** Early Nok potsherds from the Doguwa site (DOG 3_8008, DOG 3_8054), belonging to the Puntun Dutse pottery group. **2** Early Middle Nok potsherd from the Ifana site (IFA 3_2714) belonging to the Ifana pottery group. **3** Later Middle to Late Nok potsherds from the Ungwar Kura site (UK 9_1738) belonging to the Ungwar Kura pottery group.

15 typical 'beeswax' profiles at concentrations <100 µg g⁻¹ and 10 potsherds with concentrations >100 µg g⁻¹ (Table 2, shown in bold). These lipid profiles suggest the presence of beeswax in the vessels[36,37] and thus 19 of the sherds were selected for further analysis by solvent extraction to identify higher molecular weight compounds, wax esters (fatty acyl monoesters) and hydroxy wax esters (hydroxyl fatty acid monoesters), which would unambiguously confirm the presence of beeswax in the sherds.

Concentrations in the remaining six vessels were deemed too low to produce meaningful results from further analysis.

Following solvent extraction[32], five profiles (NOK003, NOK081, NOK119, NOK120 and NOK167) comprised series of higher molecular weight compounds denoting the presence of beeswax, with the remainder likely being at too low concentration for preservation of the higher molecular weight compounds (see Table 2 and Figs. 4 and 5). A further vessel, NOK376, also

**Table 1 Total number of potsherds (from each category and period) sampled for lipid analysis, shown by group, and number of potsherds containing lipids, as percentage of each group.**

| Category | Time period | Total potsherds | % of total sherds by group | Lipid-yielding sherds | % of lipid-yielding sherds by category |
|---|---|---|---|---|---|
| Early Nok (EN) | 15th–11th century BC | 150 | 33 | 20 | 13 |
| Transition (TR) | 10th century BC | 22 | 5 | 3 | 14 |
| Early Middle Nok (EMN) | 9th–8th century BC | 105 | 23 | 24 | 23 |
| Later Middle Nok (LMN) | 8th–5th century BC | 82 | 18 | 13 | 16 |
| Common Era (CE) | First and second millennium AD | 76 | 16 | 5 | 7 |
| Unknown (UNK) | Not known | 23 | 5 | 1 | 4 |
| Total | | 458 | 100 | 66 | |

contained wax esters, albeit in low abundance. Compounds dominating in these lipid profiles included the $C_{28}$, $C_{30}$ and $C_{32}$ $n$-alkanols and $C_{24}$ and $C_{26}$ $n$-alkanoic acids, and, eluting at longer retention times, were a series of $C_{40}$–$C_{52}$ carbon number palmitic acid wax esters, maximising at $C_{46}$. A series of hydroxy palmitic acid wax esters, eluting at somewhat longer retention times than the wax esters, in which the $C_{48}$ and $C_{50}$ homologues were the most abundant components, are also present in samples NOK003, NOK081, NOK119, NOK120 and NOK167. These results unambiguously confirm these vessels contained beeswax residues. A further eight potsherds (NOK15, NOK93, NOK106, NOK127, NOK158, NOK246, NOK300 and NOK376, Table 2) also yielded lipid extracts containing the characteristic distributions of both $n$-alcohols ($C_{22}$–$C_{34}$) and $n$-alkanes ($C_{21}$–$C_{31}$), described above. These lipid profiles resembled those of the six sherds described above but lacked the intact wax esters and hydroxy palmitic acid wax esters present in these sherds.

**Modern beeswax lipid distributions**. Fresh beeswax comprises a complex mixture of aliphatic compounds which consist of series of homologues differing in chain-length by two methylene groups. The $n$-alkanoic acids range from $C_{20}$ to $C_{36}$ (usually dominated by lignoceric acid, $C_{24}$) and the medium-chain $n$-alkanes range from $C_{23}$ to $C_{31}$ (with $C_{27}$ dominating in A. mellifera). Monoesters comprise predominantly alkyl palmitates ($C_{38}$–$C_{52}$), with characteristic hydroxy monoesters comprising long-chain alcohols ($C_{24}$–$C_{38}$) esterified mainly to hydroxypalmitic acid[38], ranging between $C_{40}$ and $C_{54}$. Whilst the chromatographic profile of ancient beeswax is relatively resistant to degradation, it often presents differences to that of modern beeswax[12,37]. It is known the free $n$-alkanols are not seen in fresh beeswax but do occur in aged wax, resulting from hydrolysis of the wax esters. Furthermore, a preferential loss of shorter chain $n$-alkanes may result in changes to the $n$-alkane profile across time[39].

**African honeybee species**. There are 10 species of Apis, which, aside from A. mellifera, are confined to Asia. The native distribution of the western honeybee (A. mellifera L.) encompasses Africa, Europe and western Asia[40,41]. In Africa, there are 10 subspecies of A. mellifera[42], with A. mellifera adansonii being regarded as the 'indigenous' Western Africa subspecies[43,44], making it the likely candidate for the Nok beeswax lipids. Its distribution area of the wet tropical and equatorial zone along the coast of West Africa overlaps with that of A. mellifera jemenitica subspecies to the north, who occupy the Sahel and the drier savannas of the North Sudan vegetation zone and some hybridisation between the two subspecies is thought to occur[42]. Significantly, it has been shown that there are no notable differences in the basic chemical composition of wax originating from

different A. mellifera subspecies, only small variations related to the proportion of the predominant compounds[45–47], i.e. fatty acid esters (~67%), hydrocarbons (~14%), and free fatty acids (~13%). Stingless bees (Meliponines) are also exploited in Africa, although their honey yield is known to be much lower[2].

**Beeswax lipid-yielding vessels**. Vessel NOK119 (Figs. 4 and 5), dating to the Early Nok period, displaying a typical beeswax profile including wax esters and hydroxyl wax esters, also contained the major fatty acids, $C_{16:0}$ and $C_{18:0}$, typical of animal product processing, in relatively high abundance. Furthermore, the lipid concentration of this vessel was the highest of all possible beeswax residue-bearing vessels at $1.8\ \mathrm{mg\,g^{-1}}$, suggesting that beeswax or, possibly, honey may have been mixed with animal products in this vessel, similarly to the mixing of fats and waxes in two Late Saxon/early medieval ceramic vessels[48] recovered from West Cotton, Northamptonshire, UK. A further two vessels (NOK167, Early Nok and NOK376, Middle Nok) also displayed relatively high concentrations at 0.9 and $0.6\ \mathrm{mg\,g^{-1}}$, respectively. Of these two vessels, NOK167, with a lipid profile including wax esters ($C_{40}$–$C_{48}$) and hydroxy wax esters ($C_{46}$–$C_{48}$), also contained high abundances of the $C_{16}$ fatty acid and lower abundances of the $C_{18}$ again suggesting these vessels may have been used in the processing of bee and animal carcass products. Following solvent extraction, vessel NOK376 did not include $C_{16:0}$ and $C_{18:0}$ fatty acids, typical of animal product processing, and yielded very low abundances of wax esters and was thus likely to have been used solely for bee product processing.

The 25 potsherds containing compounds suggestive of the presence of beeswax come from three periods: Early Nok, Early Middle and Later Middle Nok, perhaps not surprisingly, as these categories comprised the largest numbers of sherds sampled and lipid-containing sherds.

These data demonstrate that beeswax, a direct indicator of bee exploitation, occurs throughout the Nok culture. Interestingly, no biomarkers for beeswax were found in (later) Common Era sherds. Indeed, only 8% ($n = 5$, Table 1) of these sherds yielded lipids suggesting either that, overall, Common Era pottery was not used as much or lipid preservation conditions were less favourable. Overall, beeswax occurred in 38% of lipid-yielding sherds across all periods but was the most frequent commodity processed in Early Nok pottery (55%, $n = 11$, Table 2), decreasing somewhat in Early and Later Middle Nok pottery (38% and 38%, $n = 9$ and $n = 5$, respectively, Table 2).

**Discussion**

Where available, honeybee hive products, including honey, beeswax and brood (pupae and larvae) would likely have been of considerable importance to ancient communities, both as a nutritional source and also for medicinal, cosmetic and

**Table 2 List of 25 Nok potsherds showing 'typical' beeswax lipid profiles.**

| Sample number | Pottery group (Phase) | Site | Sherd type | Acid extraction | Solvent extraction & HT-GC/MS | Lipid concentration ($\mu g\ g^{-1}$) | Results |
|---|---|---|---|---|---|---|---|
| NOK003 | Tsaunim Gurara (Later Middle Nok) | Tudun Dosa | Rim | ✓ | ✓ | 31.8 | FA, OH, ALK, HWE, beeswax |
| NOK012 | Puntun Dutse (Early Nok) | Doguwa | Rim | ✓ | – | 26.6 | FA, OH, ALK, possible beeswax but concentration too low for solvent extraction |
| NOK013 | Puntun Dutse (Early Nok) | Doguwa | Rim | ✓ | ✓ | 47.5 | FA, OH, ALK, but WE or HWE not present, possible beeswax |
| NOK015 | Puntun Dutse (Early Nok) | Doguwa | Rim | ✓ | ✓ | **126.0** | FA, OH, ALK, but WE or HWE not present, possible beeswax |
| NOK042 | Tsaunim Gurara (Later Middle Nok) | Akura | Rim | ✓ | – | 93.5 | FA, OH, ALK, possible beeswax but contained high abundances of plasticisers so was not re-extracted |
| NOK074 | Ifana (Early Middle Nok) | Ifana | Rim | ✓ | – | 14.0 | FA, OH, ALK, possible beeswax but concentration too low for solvent extraction |
| NOK081 | Ifana/Pangwari (Early Middle Nok) | Ifana | Body | ✓ | ✓ | 54.3 | FA, OH, ALK, WE, HWE, beeswax |
| NOK084 | Ifana/Pangwari (Early Middle Nok) | Ifana | Neck | ✓ | – | 14.7 | FA, OH, ALK, possible beeswax but concentration too low for solvent extraction |
| NOK086 | Ifana/Pangwari (Early Middle Nok) | Ifana | Rim | ✓ | – | 23.2 | FA, OH, ALK, possible beeswax but concentration too low for solvent extraction |
| NOK093 | Ifana (Early Middle Nok) | Ifana | Rim | ✓ | ✓ | **170.4** | FA, OH, ALK, but WE or HWE not present, possible beeswax |
| NOK106 | Puntun Dutse (Early Nok) | Ankoro | Body | ✓ | ✓ | **213.6** | FA, OH, ALK, but WE or HWE not present, possible beeswax |
| NOK111 | Ifana (Early Middle Nok) | Daji Gwana | Neck | ✓ | ✓ | 76.9 | FA, OH, ALK, but WE or HWE not present, possible beeswax |
| NOK119 | Puntun Dutse (Early Nok) | Doguwa | Body | ✓ | ✓ | **1815.6** | FA, OH, ALK, WE, HWE, beeswax |
| NOK120 | Puntun Dutse (Early Nok) | Doguwa | Neck | ✓ | ✓ | **156.3** | FA, OH, ALK, WE, beeswax |

**Table 2 (continued)**

| Sample number | Pottery group (Phase) | Site | Sherd type | Acid extraction | Solvent extraction & HT-GC/MS | Lipid concentration ($\mu$g g$^{-1}$) | Results |
|---|---|---|---|---|---|---|---|
| NOK127 | Puntun Dutse (Early Nok) | Doguwa | Neck | ✓ | ✓ | 91.5 | FA, OH, ALK, but WE or HWE not present, possible beeswax |
| NOK130 | Puntun Dutse (Early Nok) | Doguwa | Body | ✓ | ✓ | 46.1 | FA, OH, ALK, but WE or HWE not present, possible beeswax |
| NOK158 | Puntun Dutse (Early Nok) | Puntun Dutse | Body | ✓ | ✓ | **202.1** | FA, OH, ALK, but WE or HWE not present, possible beeswax |
| NOK167 | Puntun Dutse (Early Nok) | Puntun Dutse | Body | ✓ | ✓ | **863.7** | FA, OH, ALK, WE, HWE, beeswax |
| NOK195 | Puntun Dutse (Early Nok) | Puntun Dutse | Neck | ✓ | ✓ | 59.5 | FA, OH, but WE or HWE not present, possible beeswax (or plant) |
| NOK246 | Ifana/Pangwari (Early Middle Nok) | Ifana | Body | ✓ | ✓ | **139.4** | FA, OH, ALK, but WE or HWE not present, possible beeswax |
| NOK275 | Ungwar Kura (Later Middle Nok) | Ungwar Kura | Body | ✓ | ✓ | 78.6 | FA, ALK, but WE or HWE not present, possible beeswax (or plant) |
| NOK276 | Ungwar Kura (Later Middle Nok) | Ungwar Kura | Body | ✓ | – | 32.4 | FA, OH, ALK, possible beeswax but concentration too low for solvent extraction |
| NOK300 | Ungwar Kura (Later Middle Nok) | Ungwar Kura | Body | ✓ | ✓ | **124.9** | FA, OH, ALK, but WE or HWE not present, possible beeswax |
| NOK376 | Ifana (Early Middle Nok) | Janruwa A | Body | ✓ | ✓ | **577.8** | FA, OH, ALK, WE present in very low abundance, probable beeswax |
| NOK397 | Ifana/Pangwari (Early Middle Nok) | Janruwa A | Rim | ✓ | ✓ | 77.0 | FA, OH, ALK, but WE or HWE not present, possible beeswax |

Sample number, Nok phase, site, sherd type, extraction methods and HT-GC/MS, lipid concentration ($\mu$g g$^{-1}$) and results showing presence (or absence) of 'typical' compounds found in beeswax. Potsherds containing lipids concentrations $\geq$100 $\mu$g g$^{-1}$ are shown in bold. FA *n*-alkanoic acids, OH *n*-alkanols, ALK *n*-alkanes, WE fatty acyl monoesters, HWE hydroxyl fatty acid monoesters.

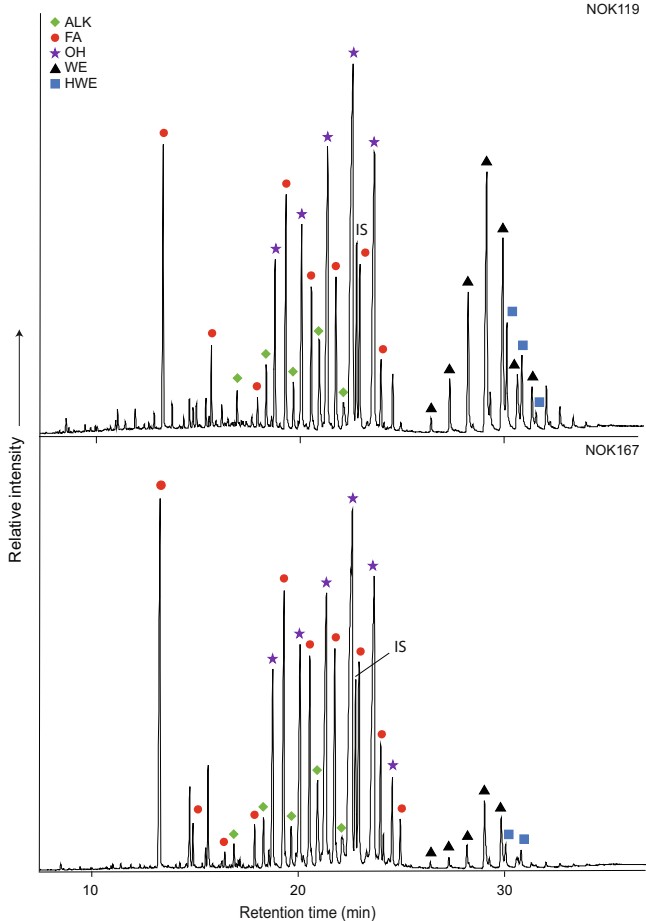

**Fig. 4 High-temperature gas chromatograms showing beeswax lipid distributions.** Partial high-temperature gas chromatograms of trimethylsilylated TLEs from Nok pottery extracts NOK119 and NOK167, red circles, *n*-alkanoic acids (fatty acids, FA); green rhombus, *n*-alkanes (ALK); purple star, *n*-alkanols (OH); black triangles, fatty acyl monoesters (WE); blue squares, hydroxyl fatty acyl monoesters (HWE) and IS internal standard, $C_{34}$ *n*-tetratriacontane.

technological purposes. Honey is the most important insect-related food globally[2] and, as a rare source of sweetener, comprises 80–95% sugar (carbohydrate), several essential vitamins and minerals and components that act as preservatives[4,49–51]. In addition, bee brood are a good source of protein, fat, several essential minerals, and B-vitamins[52,53] and are today used as a food source in many places, including some African countries, China, Mexico and Thailand[54–58]. Bee products, including honey, propolis, royal jelly and venom, possess various bioactive properties and have a history of use for various medicinal purposes, both in West Africa and globally. For example, propolis has both antiseptic and anaesthetic properties and is often used as an ingredient in medicines, toothpastes, oral sprays and chewing gums and royal jelly is valued as a medicine, tonic or aphrodisiac, likely because it contains many insect growth hormones[15].

Beeswax itself has been used for technological purposes since the Palaeolithic, with the earliest known use being as an adhesive at Border Cave, South Africa, ca. 40,000 years ago, where a lump of carefully curated organic material, comprising a mixture of beeswax and *Euphorbia tirucalli* resin wrapped in vegetal fibres, was likely used to haft a bone point[59]. Beeswax has also variously been used from prehistoric times as a sealant or waterproofing agent on Early Neolithic collared flasks in northern Europe[60], as a lamp illuminant in Minoan Crete[39] and mixed with tallow, possibly for making candles, in medieval vessels at West Cotton, Northamptonshire[48].

The presence of beeswax in ancient pottery, identified through the complex lipid distributions discussed previously, most likely arises as a consequence either of the processing (melting) of wax combs through gentle heating, leading to its absorption within the vessel walls, or, alternatively, beeswax is assumed to act as a proxy for the processing (cooking) or storage of honey itself[12]. The presence of high lipid concentrations in some Nok vessels suggests they may have been used in cooking or heating honey, possibly as an additive to other dishes, or storing it for consumption. Where honey is available, it is often an important food source for hunter-gatherers[2] and there are several groups in Africa, such as the Efe foragers of the Ituri Forest, Eastern Zaire, who have historically relied on honey as their main source of food, collecting all parts of the hive, including honey, pollen and bee larvae, from tree hollows which can be up to 30 m from the

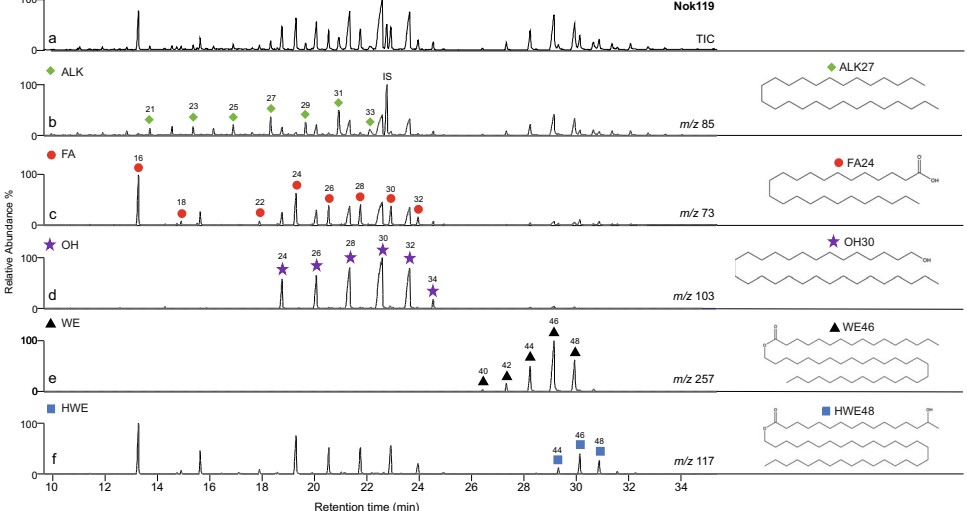

**Fig. 5 High-temperature gas chromatography/mass spectrometry chromatograms of a total lipid extract, containing beeswax, from potsherd NOK119.** **a**. partial total ion current chromatogram and **b**-**f**. mass chromatograms displaying ion masses of characteristic fragments from the main compound classes comprising the extract (*m/z* 85, 73, 103, 257 and 117, respectively), showing one of the molecular structures of each compound class. Green rhombus, *n*-alkanes (ALK); red circles, *n*-alkanoic acids (fatty acids, FA); purple stars, *n*-alkanols (OH); black triangles, fatty acyl monoesters (WE); blue squares, hydroxyl fatty acyl monoesters (HWE); IS, internal standard (*n*-tetratriacontane). Numbers denote carbon chain length.

ground, using smoke to distract the stinging bees[17]. The Hadza people, a forager group living in northern Tanzania, also rank honey as their favourite food[2,61] which they mainly collect whilst climbing Baobab trees[2]. Honey also makes other products more storable. Among the Okiek people of Kenya, who rely on the trapping and hunting of a wide variety of game, smoked meat is preserved with honey, being kept for up to 3 years[13], hence, Nok vessels which contained a mixture of animal fats and beeswax in high abundances, could have been used as containers to preserve meat in honey. The Okiek people, who sometimes use leather bags to ferment honey, keep separate pots for cooking and honey use, which are generally the same size but of a different form, with the honey pots having a narrower and longer neck than the cooking pots[62].

The importance of bee larvae and pupae to modern foragers is also well-attested[4,16,17], suggesting a further possible use of Nok vessels was to heat the combs containing brood. Once the beeswax was melted, it could be drained off leaving the larvae and pupae, which, as mentioned, are excellent sources of protein and fat. Today, Hadza foragers do not remove the bee larvae from the combs as they are eaten, and this provides an important fat source during parts of the year[4], particularly in the rainy season when hunting is less productive, the time when bee products are at their most available[2].

Honey and other bee products are also an important resource for modern West African farming populations who provide hives for feral bees to colonise[63,64]. The first farmers appear in central Nigeria (likely from the North) at around 1500 BC and may have gained their knowledge of bee behaviours through cultural contacts with indigenous hunter-gatherers, similarly to the adoption of local *Canarium schweinfurthii* fruits, a valuable source of fat[31]. Together with small-scale farming, Nok groups are known to have exploited wild resources[31].

Given the importance of honey in making honey-based drinks, wine, beer and non-alcoholic beverages, in Africa today, the Nok vessels may have been used for this purpose, although it should be noted that honey has not been identified and the chemical identification of ancient fermentation is notoriously difficult[65]. Honey may have been one of the first foodstuffs used by ancient humans to produce alcoholic beverages[10,66] although the antiquity of human alcohol production and consumption is not known. Nonetheless, there are a vast range of indigenous alcoholic beverages produced across Africa, by at least 50 different groups, including those made from honey[14,66]. Ethnographic and historical accounts of the production of honey-based drinks in Africa confirm that honey would normally be diluted in water to produce drinks, as described by Ibn Battuta, visiting the town of Walata, Mauritania, in 1352, who describes a sour drink made from ground millet mixed with honey and sour milk, with water added, served at a reception for merchants from the Maghreb[67]. The production and consumption of honey-based alcoholic drinks by prehistoric groups in Africa would likely have been valuable in creating and maintaining social, economic and political relations, possibly through the performance of ritual and ceremonial practices[68].

A further possibility is that the pots themselves may have been used as beehives (Fig. 6a–c), implying some management of the bees. This is commonplace today in East and South Africa and also evidenced in modern-day Nigerian traditional beekeeping or bee maintaining[15]. Here, pottery hives are either placed on the ground or in trees, while other types of hives made from clay, mud, straw or bark are always placed in trees[63] (Fig. 6d). Little, or no, management of the hives takes place, in contrast to modern

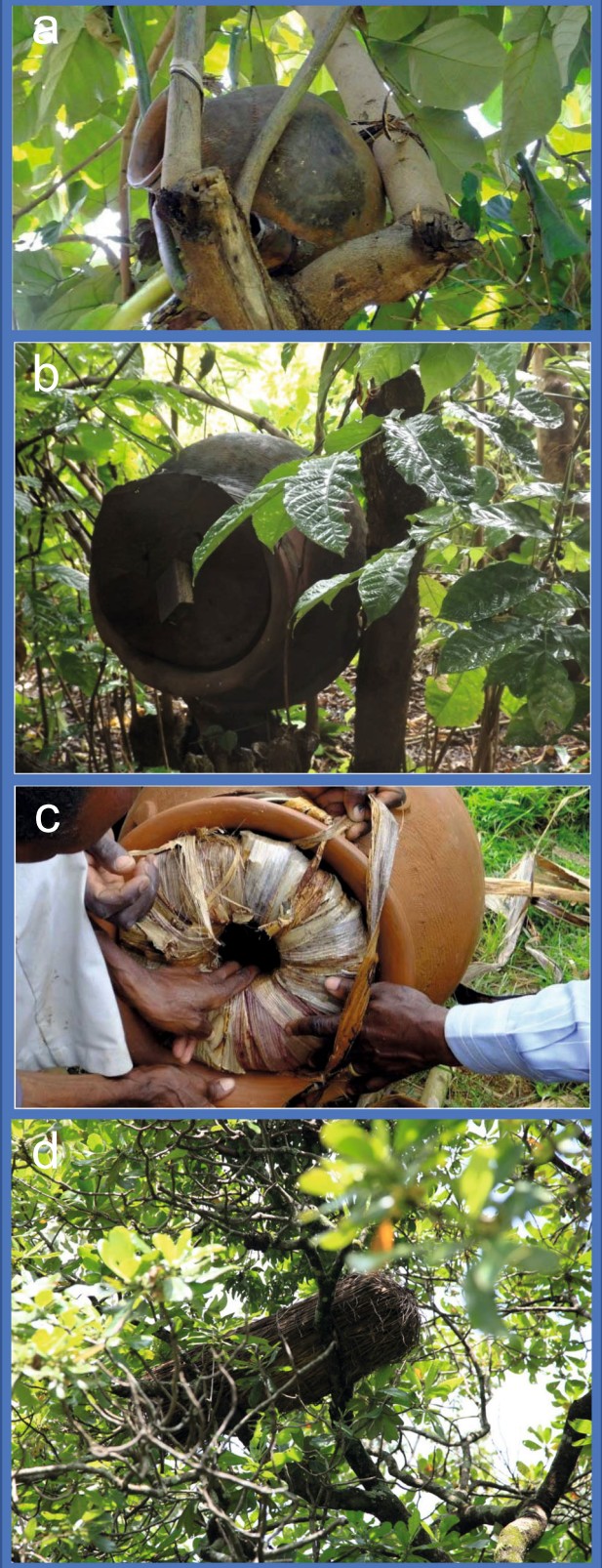

**Fig. 6 Examples of modern-day African beehives. a–c** Images of ceramic pots from Uganda, set in trees, being used as hives (images courtesy of Bees for Development) and **d** image of straw hive in a tree in Nigeria.

European beekeeping, mainly because of the tendency to abscond in tropical African honeybees. Once the colony has been in place for several months, harvesting of the combs takes place, sometimes destroying the nest. This usually takes place early in the rainy season, once the bees have exploited the freshly blossoming trees and before the heavy rains restrict the colony from collecting nectar and pollen[4266].

Clay hives have been recorded in Nigeria, Burkina Faso, Malawi and Ethiopia[63,66,69] and clay vessels with holes in them were used as beehives in Mozambique until the 1970s and are still used in Kenya and Uganda today[69,70]. However, Nok vessels are generally only 20–30 cm diameter, and likely too small to have been used for these purposes.

Finally, although honey can be squeezed from the bee combs by hand or sieved through a mesh, the invention of pottery represents a further stage of food processing for human groups, in this instance, allowing controlled heating of the comb to separate the wax, honey and/or larvae. Given the early use of pottery in prehistoric West Africa, this first identification of beeswax residues, whether through honey collecting or beekeeping, in an early farming context 3500 years ago, hints at a much older history of bee exploitation. Further lipid residue analysis could confirm whether forager/honey-hunting groups exploited honeybees for their bee products some 8000 years earlier, when pottery was first invented in Africa.

## Methods

**Lipid extraction**. Lipid analysis and interpretations were performed as detailed in previous publications[32,33]. The reagents used were analytical grade (typically > 98% of purity) and solvents were of HPLC grade (Rathburn). In brief, for each sample, around 2 g of cleaned potsherd was crushed and transferred into furnaced culture tubes. A known amount of internal standard was added (20 μg $n$-tetratriacontane; Sigma Aldrich Company Ltd.) and the lipids were then esterified/transesterified using 5 mL of $H_2SO_4$/MeOH 2–4%, by heating for 1 h at 70 °C, mixing every 10 min. Following this, the extract was centrifuged at $660 \times g$ for 10 min. The supernatant was then removed to a clean culture tube and 2 mL of DCM extracted double-distilled water was added. In order to recover any lipids not fully solubilised by the methanol solution, $2 \times 3$ mL of $n$-hexane was added to the extracted potsherds contained in the original culture tubes, mixed well and transferred to a second culture tube. A further $2 \times 2$ mL $n$-hexane was then added directly to the $H_2SO_4$/MeOH solution and whirlimixed to extract any remaining lipids, and then combined in 3.5 mL vials and blown down until a full vial of the TLE remained. An aliquot of each TLE was derivatised using $N,O$-bis(trimethylsilyl)trifluoroacetamide (BSTFA) containing 1% trimethylchlorosilane (TMCS; Sigma Aldrich Company Ltd., 20 μL; 70 °C, 1 h), excess BSTFA was removed under a gentle stream of nitrogen in a heating block at 40 °C and the derivatised TLE was dissolved in $n$-hexane prior to analysis by GC and GC–MS.

Further analysis was carried out using solvent extraction of cleaned and crushed potsherds. An internal standard was added to the sherd powder ($n$-tetratriacontane, typically 20 μg), to enable lipid quantification, and they were solvent extracted by ultrasonication (chloroform/methanol 2:1$v/v$, 30 min, $2 \times 10$ mL). After centrifugation, the solvent was decanted into 3.5 mL vials and blown down to dryness using a gentle stream of nitrogen, leaving the TLE. Aliquots of the TLE were filtered through a silica column and trimethylsilylated using $N,O$-bis(trimethylsilyl)trifluoroacetamide, 20 μL, 70 °C, 1 h, followed by dilution with $n$-hexane and analysis by high-temperature-GC (HTGC) and HTGC-MS.

**Instrumental analysis**. Analyses of the TLEs was carried out on an Agilent 7820A gas chromatograph, using on-column injection with the flame ionisation detector (FID) set to 300 °C, and fitted with a high-temperature non-polar column (DB1-HT; 100% dimethylpolysiloxane, 15 m × 0.32 mm i.d., 0.1 μm film thickness). The carrier gas was helium, set to a constant flow of 2 mL min$^{-1}$ and the temperature programme comprised a 50 °C isothermal hold followed by a gradient increase to 350 °C at a rate of 10 °C min$^{-1}$ followed by a 10 min isothermal hold. Blanks (no sample) were prepared and analysed alongside every batch. Data was acquired using HP Chemstation software (Rev. C.01.07 (27), Agilent Technologies). Further compound identification was accomplished using high-temperature gas chromatography–mass spectrometry (HTGC–MS). FAMEs were introduced by autosampler onto a ThermoScientific Trace 1300 gas chromatograph directly coupled to an ISQ single quadrupole mass spectrometer, fitted with a non-polar column, 15 m × 0.32 mm fused silica capillary column coated with a stationary phase (100% dimethylpolysiloxane, Restek, 0.17 μm). The initial injection port temperature was 50 °C with an evaporation phase of 0.05 min, followed by a transfer phase from 50 °C to 380 °C at 0.2 °C min$^{-1}$. The oven temperature was held isothermally for 2 min at 50 °C, increasing by 10 °C min$^{-1}$ to 280 °C, then at a rate of 25°C min$^{-1}$ to 380 °C with a final hold at 380 °C for 5 min. Helium was used as a carrier gas and maintained at a constant flow 5 mL min$^{-1}$. Operating conditions were as follows: electron ionisation (EI) mode (70 eV) with a GC interface temperature of 380 °C, source temperature 340 °C and emission current of 50 μA. The instrument was set to acquire in the range of $m/z$ 50–950 Da at two scans s$^{-1}$ in full scan mode.

Data acquisition and processing was carried out using Xcalibur software (version 3.0). Peaks were identified on the basis of their mass spectra and GC retention times, by comparison with the NIST mass spectral library (version 2.0) and modern beeswax (from the Loire department, France).

**Reporting summary**. Further information on research design is available in the Nature Research Reporting Summary linked to this article.

## Data availability
The authors declare that all data supporting the findings of this study are available within the paper.

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

## Acknowledgements

The authors wish to thank NERC 771 (Reference: CC010) and NEIF (www.isotopesuk.org) for funding and maintenance of the instruments used for this work and Ian Bull, Alison Kuhl and Helen Whelton for technical help. We especially thank the Deutsche Forschungsgemeinschaft for the financial funding of this project (BR 1459/7 and NE 408/5) and the National Commission for Museums and Monuments in Nigeria for aiding us in conducting research on the Nok Culture and providing staff to assist in the fieldwork.

## Author contributions

J.D., K.N. and R.P.E. conceived the project. J.D., T.G. and C.W.-D. performed the experimental work and GC–MS analyses. G.F. assessed pottery characteristics and provided photographs. G.F. and J.D. selected the samples. K.N., P.B., G.F. and A.H. provided insights into excavation contexts and archaeobotanical insight. J.D., A.H. and G.F. wrote the manuscript, with contributions from all authors.

## Funding

## Competing interests

The authors declare no competing interests.
