## [Peer Review File · Nature Communications]

Reviewers' Comments:

Reviewer #1:

Remarks to the Author:

Review report on:

Manuscript Number: NCOMMS – 20 – 31945

Title: Honey hunting in prehistoric West Africa 3500 years ago

- What are the major claims of the paper? /What are the noteworthy results?

The paper provides the first direct chemical evidence of beeswax/honey processing in West African ceramic vessels, and thus honey-hunting, in an early farming context, around 3500 years ago. Additionally, the study provides the first biomolecular evidence for the 115 palaeoecological range of *Apis mellifera adansonii* in Holocene West Africa. The study also claims that the testimonies of historical and ethnographic literature bear direct relevance to testimonies of archaeology.

- Are the claims novel?

Yes. It is a pioneer study of significant proportions in the Holocene archaeology of West Africa and sub-Saharan Africa in general. The reliability of the testimonies of historical (mostly oral sources) and ethnographic data have been proven by archaeological data from this study to extend to at least about 3,500 years. This has profound implications for the reconstruction of the past of 'non-literate' peoples as it has paved the way for the celebration of historical (most especially oral traditions) and ethnographic sources as potentially authentic data for the reconstruction of early human activities in sub-Saharan Africa and by extension among what has often been referred to 'non-literate' societies.

- Will the paper be of interest to others in the field? / Will the work be of significance to the field and related fields?

Yes. The paper will foster the much needed awareness concerning the much less known (in comparison to other parts of Africa) Holocene food producing activities of early humans in West Africa.

- One area that might engender interests is in the area of the complex interaction between anthropogenic activities and climate change during Holocene West Africa. the impact of these on global climate change and how these can thus contribute to efforts at mitigating global climate change challenges.

- Will the paper influence thinking in the field?

Yes. The paper is capable of enlisting further information that can aid a better understanding of the very little known but complex anthropogenic activities in West Africa during the Holocene period.

- Are these claims convincing?

Yes, very convincing. Except for hints in Paleolithic arts this aspect of food production (honey and bee wax hunting) is seldom encountered in archaeological contexts.

- Are there other experiments that would strengthen the paper further?

No

- Are the claims appropriately discussed in the context of previous literature?

Yes, the paper adequately interrogated relevant literature in its claims.

- If the manuscript is unacceptable in its present form, does the study seem sufficiently promising that the authors should be encouraged to consider a resubmission in the future

The paper is suitable for publication in its present form.

- Is the manuscript clearly written?

Yes, the language and expressions used are very clear.

- Could the manuscript be shortened to aid communication of the most important findings?

The manuscript is concise enough in its present form, to allow for significant information necessary for clarity to be retained.

- Have the authors done justice with overselling their claims?

Yes. They were able to situate their claims within the context of the state of knowledge in this particular area of study thus clearly highlighting the gaps their research has been able to fill.

- Are there any flaws in the data analysis, interpretations and conclusions? – Does these prohibit publication or require revision?

No

- Does the work support the conclusions and claims, or is additional evidence needed?

Yes, there is unity between the work and conclusions and claims. Additional evidence is not needed.

- Have they been fair in their treatment of existing literature? /How does it compare to the established literature?

Yes, existing literature interrogated were up to date (i.e. year 2020).

- Have they provided sufficient methodological detail that the experiments could be reproduced? / Is there enough detail provided in the methods for the work to be reproduced?

Yes, the methodological procedures were clearly enumerated for replication.

- Is the methodology sound?

Yes

- Does the work meet the expected standards in your field?

Yes

- Is the statistical analysis of the data sound?

Yes, the exact sample size (n) for each experimental group/condition were given as discrete number and unit of measurement. Additionally, a statement on whether measurements were taken from distinct samples or whether the same sample was measured repeatedly was confirmed.

- Should the authors be asked to provide further data or methodological information to help others to replicate their work (such data might include source code for modelling studies, detailed protocols or mathematical derivations)?

No

- Are there any specific ethical concerns from the use of animals or human subjects?

No, n/a.

There are minor suggested edits in lines 90 and 333 which may be taken into consideration.

Prof. Jonathan Oluyori Aleru

Reviewer #2:

Remarks to the Author:

In this paper, the authors have analysed 458 potsherds from 12 Nok archaeological sites from Nigeria. A total of 66 of them (14,4 %) provided significant amount of lipids. The molecular assemblages detected and identified by HTGC and HT-GCMS allow the distinction of three categories of content: animal fats, various plant residues and beeswax. This article is focused on this latter natural substance.

The paper is very well organised and written and it provides new and valuable information on the exploitation of beehive products for a chrono-cultural context for which very few data are available on the subject.

The methods are well described and adapted to the purpose of the work.

The conclusions are well supported by the data and the discussion deals with the different aspects of beehive products uses.

I am therefore very much in favor of the publication of this article in Nature communications subject to minor revisions on the beeswax/honey aspects and on the issue of harvesting beekeeping substances by honey hunting. I also have some other questions that I mention below.

The article speaks indifferently of honey and beeswax: "beeswax/honey". In the abstract, one can read « with direct chemical evidence for beeswax and honey ». If there is a lot of chemical evidence for beeswax, this is not true for honey that is not well preserved. As Eva Crane wrote it, « Wherever social bees exist, man exploits them for their honey by hunting or beekeeping. » Nevertheless, the chemical evidence found in ceramic vessels is that of beeswax and not of honey. This is well explained on page 14 by the sentence « beeswax is assumed to act as a proxy for the processing (cooking) or storage of honey itself ». So, I think that, before this sentence, the expression « beeswax/honey » must not be employed and must be replaced by the only product chemically identified in the potsherds « beeswax ».

p. 6, lines 112-113: « Hence, we provide the first direct chemical evidence of beeswax/honey processing in West African ceramic vessels, and thus honey-hunting, in an early farming context, around 3500 years ago, and, additionally, the first biomolecular evidence for the palaeoecological range of *Apis mellifera adansonii* in Holocene West Africa. » I think that the conclusion « and thus honey-hunting » has to be developed. What argument allows the authors to hypothesize that this is honey hunting and not beekeeping, even proto-beekeeping? This should be discussed more thoroughly.

How can authors be sure that the beeswax identified was produced from *Apis mellifera adansonii*? This is discussed p. 13 but I am not sure that it is possible to be so precise.

p. 16, lines 293-294: « Finally, the invention of pottery represents a further stage of food processing for human groups, in this instance, allowing controlled heating of the comb to separate the wax and honey. » Ceramic vessels are not essential to separate beeswax from honey. For instance, the film of Eric Valli and Diane Summers on honey hunters of Nepal shows people who separate these two substances with their hands. This film could be mentioned and discussed in the paper for the practices without pottery.

And a last thing: in the abstract as well as on pages 7 and 8, the authors indicate that they analysed 458 potsherds but page 6 talks about 450 potsherds. I guess that this is a mistake and that 450 has to be replaced by 458.

Reviewer #3:

Remarks to the Author:

This study applies a well-established methodology to pottery sherds belonging to the Nok culture of West Africa. A total of 458 potsherds from 12 sites were investigated. Lipids were only recovered from < 20% of the sherds. Of this group, 25 lipid profiles are consistent with beeswax, although the quality and abundance of the biomarker distributions is variable. These results are discussed in the context of beeswax and honey use.

Beeswax has been identified in archaeological contexts using GC-MS on numerous occasions over the past 25 years. The methodology is routine as is the chemical identification of beeswax. Mixtures of beeswax with other lipids is explored. Much of the discussion draws on documentary evidence for the diverse uses of honey and beeswax in West Africa. The analytical data presented here simply provides a range of possibilities in terms of wider use and roles of beeswax and honey among these communities. There is no information as to whether these roles changed over the 1500 year period of the Nok culture.

I am happy to recommend publication in Nature Communications, subject to the following points that the authors should consider:

(i) Beeswax/honey. To avoid confusion, the authors should distinguish more carefully the identification of beeswax, reliably presented in the manuscript, and the presumed presence of honey, which due to its sugar content will rarely survive in archaeological contexts. There is an example on page 10/line 171 (These results unambiguously confirm these vessels contained beeswax/honey residues). Simply put, the results confirm the former but not the latter. There are many examples from other contexts where beeswax was used as a product in its own right. There is a comment in lines 251-252 but greater clarity is needed overall.

(ii) Chronology. The Nok culture spans 1500 years. Table 1 breaks down this down into five consecutive periods. For completion, can the authors include the dates of each of the chronological periods?

(iii) Pottery vessels. The authors do not include any information of the form/type of the vessels, especially if there is a relationship between form and residue type. Can the authors provide more information here? Could some of the pottery vessels have served as beehives?

(iv) Uses of wax and honey. The authors should include information, pertinent to the West African context, on the medicinal uses of honey and wax. The authors don't mention propolis, which might also be included here. The text is rather dominated by relationships to food and drink.

Reviewer #4:

Remarks to the Author:

This manuscript by Dunne et al. uses well-developed chromatographic methods to characterize lipids absorbed in early Iron age potsherds of the Nok culture in the humid tropics of West Africa. A total of 458 sherds was analyzed, 66 preserved lipids, beeswax was securely identified in five sherds, probable in one, and possible in 19. The identification of beeswax in pottery is not novel, in general. The main point of significance identified in the abstract is that it the first identification in West African archaeological ceramics.

This manuscript is remarkably well-constructed and would require only minor revisions for publication as a report of interest to an archaeological community. Points that could be elaborated or revised that would make this report of wider interest are discussed in the remainder of this review.

1. Honey, bee larvae and adult bees are valuable source of energy and protein exploited by many African vertebrate species, including honey badgers (*Mellivora capensis*), chimpanzees (*Pan troglodytes*, McLennan, *Am J Phys Anth* 2015; Boesch et al. *J Human Evol* 2009), humans, and avian bee-eaters (*Meropidae*). Honey guide birds (*Indicator sp.*) may digest beeswax (Downs et al., 2002. *Comparative Biochemistry and Physiology A* 133, 125-134.), and these birds actively recruit humans and honey badgers to natural beehives (Wood et al. 2014, *Evolution and Human Behavior* 35, 540). The diversity of taxa that intensively exploit African honey bees, and commensal/symbiotic relationships among species, suggests a substantial time depth of human exploitation of honey bees, honey and beeswax- beginning long before pottery and perhaps before the australopithecines. Control of fire to generate smoke to repel bees while their hives are being plundered, may have been an important inflection point in the history of human honey consumption, and also processing of wax and honey wine fermentation.

2. Pottery vessels are not a prerequisite for honey fermentation. The Okiek hunter-gatherers of highland Kenya often used leather bags for fermentation of honey, according to Blackburn.

3. Evidence for beeswax dating over 40 ka (thousand years) in Southern Africa (cited in the ms under review) and in southern Europe at approximately the same age (Sano et al. 2019. *Nature Ecology and Evolution* 3, 1409), suggests an important additional role for beeswax as a waterproofing component of gums and mastics for hafting stone tools. This is notable because most tree gums are water soluble, so glued composite artifacts could disassemble in the wet season.

4. Human biology may also reflect a deep time relationship of humans and honey wine/beer. The alcohol and aldehyde dehydrogenase alleles predominant among Africans may reflect a substantial time depth for metabolism of alcohol, including honey-based fermented beverages. In contrast, Asian populations include many individuals with alleles of both genes that make ethanol consumption uncomfortable at best.

5. The mode of acquisition of honey is referred to in the paper title, abstract and main text as "honey-hunting", and is supported by references (5-13) to collecting honey (not hunting honey) from natural hives, mainly by hunter-gatherers. Natural hives are exploited universally by African honey collectors. However, many African societies manufacture hives and are prolific beekeepers and 'cultivators', notably the Okiek hunter-gatherers of Kenya discussed in ref. 9. Traditional manufactured beehives dangle from trees almost everywhere in Sub-Saharan Africa. Ethiopian peoples have well-developed honey-based alcoholic beverage production and manufactured hives. References to beekeeping rather than honey 'hunting' are not cited until ref 62; Ref 64 refers to bee husbandry in Nigeria, and ref 65 to beekeeping in western and northern Africa. Therefore if African foragers and agriculturalists are prolific bee-keepers and honey collectors, then "hunting" is a misnomer. It diminishes the scale and technological and social complexity of well-developed African hunter-gatherer and agriculturalists honey production and exploitation systems. Moreover, it implies that the agricultural Nok peoples were not beekeepers.

6. Most interesting for the paper under review, is Irvine's review of traditional African beekeeping (1957, *Bee World* 38, 117), which describes pottery beehives that are smeared with beeswax inside to attract bee colonies. This practice could be considered one form of beeswax processing identifiable with lipid analyses. Beeswax traces in Nok sherds could also indicate pottery hives. One example of a ceramic vessel interior with traces of where the brood/honey combs were once attached is shown in fig 10 of ref 9 by Russell and Lander 2015. This kind of trace is clearly evidence for beekeeping rather than honey processing.

7. The local archaeological context could be described briefly, particularly the first appearance of pottery in this region. Mali is mentioned for the earliest pottery in Africa. Ref. 29, which I cannot access, apparently discusses evidence for millet agriculture at 3500 bp, described as contemporary with the Nok Culture. Kintampo Neolithic sites are relevant predecessors in this region. Some Kintampo sites have abundant evidence for exploitation of lipid-rich *Canarium* tree and palm nuts in the form of charred botanical remains. Lipid analysis of Kintampo ceramics might provide evidence for oil extraction or other nut processing practices, as well as earlier evidence of honey and beeswax processing in the West African savanna woodlands.

8. Figure 1 shows color variations that might represent elevation. Please explain in the caption. what is being shown with this color scheme.

9. Figure 2 shows three classic Nok effigy vessels. While these figures are impressive, it might be more relevant to show an image of a traditional hive, particularly a pottery hive. These Nok effigy ceramics would, however, be a good journal cover photo.

10. More discussion regarding how beeswax become impregnated in porous ceramics would be welcome. Many questions arise:

Does storage of strained honey without wax combs deposit wax lipids in the vessel walls?

If wax and other hydrophobic molecules have lower densities than honey, does absorption occur preferentially at the level of the liquid surface, at the vessel neck (as Evershed and others have shown for animal fats)?

If honey is brewed into mead then does the low heat used to accelerate fermentation increase lipid absorption at the vessel neck?

Does ethanol act as a solvent for some of the lipids, facilitating absorption or volatilization?

If combs were systematically melted at higher temperatures to purify the wax, does this alter the proportions of lipids of different molecular weights and volatilities?

Does wax processing concentrate lipids in the vessel bottom rather than at the neck?

Does honey have non-wax lipids or is it pure carbohydrate and water?

Do pollen and insect cuticle lipids have distinctive chromatographic signatures?

These question beg for answers through systematic experiments and analysis of ethnographically documented ceramics used for honey processing.

10. The chromatographic evidence for beeswax is not evidence for honey itself. Honey could be eaten raw from the comb, or pressed, and honey-free beeswax could then processed by melting for sealing, waterproofing, and as an additive to gums and mastics for waterproofing, among other uses.

The chromatograms in figures 4 and 5 could be more readily interpretable for this reviewer if chromatograms of beeswax reference samples were also shown, perhaps in a supplemental information section.

In summary, this paper could be improved for the general readership of Nature journals by: (1) discussing more of the evidence suggesting a considerable time depth for honey consumption in Africa, (2) discuss beekeeping practices as well as honey collecting (rather than hunting), (3) consider a broader suite of documented practices that could leave traces of beeswax in potsherds, and (4) describe how lipid types, proportions and distributions vary within vessels with different processing practices.

Reviewed by Stanley H. Ambrose

NCOMMS-20-31945

Honey hunting in prehistoric West Africa 3500 years ago.

Dunne, J^{1*}., Höhn, A^{2.}., Franke, G^{2.}., Neumann, K^{2.}., Breunig, P^{2.}., Gillard, T^{1.}., Walton-Doyle, C^{1.} and Evershed, R.P^{1.}

Responses to reviewers

Please note we have highlighted the reviewer's suggestions in bold (black text) and our responses to the reviewer's comments are inserted in red text. We have also detailed below where we have made additions to the text of the paper in red (bold).

Additions to the text of the manuscript itself are shown in red.

Reviewer #1 (Remarks to the Author):

Review report on:

Manuscript Number: NCOMMS – 20 – 31945

Title: Honey hunting in prehistoric West Africa 3500 years ago

- What are the major claims of the paper? /What are the noteworthy results?

The paper provides the first direct chemical evidence of beeswax/honey processing in West African ceramic vessels, and thus honey-hunting, in an early farming context, around 3500 years ago. Additionally, the study provides the first biomolecular evidence for the 115 palaeoecological range of *Apis mellifera adansonii* in Holocene West Africa. The study also claims that the testimonies of historical and ethnographic literature bear direct relevance to testimonies of archaeology.

- Are the claims novel?

Yes. It is a pioneer study of significant proportions in the Holocene archaeology of West Africa and sub-Saharan Africa in general. The reliability of the testimonies of historical (mostly oral sources) and ethnographic data have been proven by archaeological data from this study to extend to at least about 3,500 years. This has profound implications for the reconstruction of the past of 'non-literate' peoples as it has paved the way for the celebration of historical (most especially oral traditions) and ethnographic sources as potentially authentic data for the reconstruction of early human activities in sub-Saharan Africa and by extension among what has often been referred to 'non-literate' societies.

- Will the paper be of interest to others in the field? / Will the work be of significance to the field and related fields?

Yes. The paper will foster the much needed awareness concerning the much less known (in comparison to other parts of Africa) Holocene food producing activities of early humans in West Africa.

- One area that might engender interests is in the area of the complex interaction between anthropogenic activities and climate change during Holocene West Africa. the impact of these on global climate change and how these can thus contribute to efforts at mitigating global climate change challenges.

- Will the paper influence thinking in the field?

Yes. The paper is capable of enlisting further information that can aid a better understanding of the very little known but complex anthropogenic activities in West Africa during the Holocene period.

- Are these claims convincing?

Yes, very convincing. Except for hints in Paleolithic arts this aspect of food production (honey and bee wax hunting) is seldom encountered in archaeological contexts.

- Are there other experiments that would strengthen the paper further?

No

- Are the claims appropriately discussed in the context of previous literature?

Yes, the paper adequately interrogated relevant literature in its claims.

- If the manuscript is unacceptable in its present form, does the study seem sufficiently promising that the authors should be encouraged to consider a resubmission in the future

The paper is suitable for publication in its present form.

- Is the manuscript clearly written?

Yes, the language and expressions used are very clear.

- Could the manuscript be shortened to aid communication of the most important findings?

The manuscript is concise enough in its present form, to allow for significant information necessary for clarity to be retained.

- Have the authors done justice with overselling their claims?

Yes. They were able to situate their claims within the context of the state of knowledge in this particular area of study thus clearly highlighting the gaps their research has been able to fill.

- Are there any flaws in the data analysis, interpretations and conclusions? – Does these prohibit publication or require revision?

No

- Does the work support the conclusions and claims, or is additional evidence needed?

Yes, there is unity between the work and conclusions and claims. Additional evidence is not needed.

- Have they been fair in their treatment of existing literature? /How does it compare to the established literature?

Yes, existing literature interrogated were up to date (i.e. year 2020).

- Have they provided sufficient methodological detail that the experiments could be reproduced? / Is there enough detail provided in the methods for the work to be reproduced?

Yes, the methodological procedures were clearly enumerated for replication.

- Is the methodology sound?

Yes

- Does the work meet the expected standards in your field?

Yes

- Is the statistical analysis of the data sound?

Yes, the exact sample size (n) for each experimental group/condition were given as discrete number and unit of measurement. Additionally, a statement on whether measurements were taken from distinct samples or whether the same sample was measured repeatedly was confirmed.

- Should the authors be asked to provide further data or methodological information to help others to replicate their work (such data might include source code for modelling studies, detailed protocols or mathematical derivations)?

No

- Are there any specific ethical concerns from the use of animals or human subjects?

No, n/a.

There are minor suggested edits in lines 90 and 333 which may be taken into consideration.

Prof. Jonathan Oluyori Aleru

This reviewer had two comments (lines 90 and 333 as above).

On line 90 (which is the caption for Figure 1, map of the region and sites), the reviewer asks 'for proper reference to be provided and in the reference listing'. However, this map has been specifically made for this publication, by Eyub F. Eyub (who we are obviously crediting) and does not need to be referenced.

Line 333, we begin the sentence with the word 'firstly' and the reviewer notes that there is no second. We have removed the word 'firstly'.

Reviewer #2 (Remarks to the Author):

In this paper, the authors have analysed 458 potsherds from 12 Nok archaeological sites from Nigeria. A total of 66 of them (14,4 %) provided significant amount of lipids. The molecular assemblages detected and identified by HTGC and HT-GCMS allow the distinction of three categories of content: animal fats, various plant residues and beeswax. This article is focused on this latter natural substance.

The paper is very well organised and written and it provides new and valuable information on the exploitation of beehive products for a chrono-cultural context for which very few data are available on the subject.

The methods are well described and adapted to the purpose of the work.

The conclusions are well supported by the data and the discussion deals with the different aspects of beehive products uses.

I am therefore very much in favor of the publication of this article in Nature communications subject to minor revisions on the beeswax/honey aspects and on the issue of harvesting beekeeping substances by honey hunting. I also have some other questions that I mention below.

The article speaks indifferently of honey and beeswax: "beeswax/honey". In the abstract, one can read « with direct chemical evidence for beeswax and honey ». If there is a lot of chemical evidence for beeswax, this is not true for honey that is not well preserved. As Eva Crane wrote it, « Wherever social bees exist, man exploits them for their honey by hunting or beekeeping. » Nevertheless, the chemical evidence found in ceramic vessels is that of beeswax and not of honey. This is well explained on page 14 by the sentence « beeswax is assumed to act as a proxy for the processing (cooking) or storage of honey itself ». So, I think that, before this sentence, the expression « beeswax/honey » must not be employed and must be replaced by the only product chemically identified in the potsherds « beeswax ».

Agreed. We have either removed all reference to beeswax/honey processing to read just beeswax processing up until page 14 or have mentioned beeswax and possible honey processing, where reference to honey was needed in the text.

p. 6, lines 112-113: « Hence, we provide the first direct chemical evidence of beeswax/honey processing in West African ceramic vessels, and thus honey-hunting, in an early farming context, around 3500 years ago, and, additionally, the first biomolecular evidence for the palaeoecological range of *Apis mellifera adansonii* in Holocene West Africa. » I think that the conclusion « and thus honey-hunting » has to be developed. What argument allows the authors to hypothesize that this is honey hunting and not beekeeping, even proto-beekeeping? This should be discussed more thoroughly.

How can authors be sure that the beeswax identified was produced from *Apis mellifera adansonii*? This is discussed p. 13 but I am not sure that it is possible to be so precise.

With regard to the reviewers comment on proto-beekeeping, we agree and have made changes to the text. Please see our response to reviewer 4, who raised the same issue, for full detail on this.

With regard to our interpretation that the residues originate from *Apis mellifera adansonii* we have made this assumption because this sub-species is the 'indigenous' Western Africa subspecies (Fletcher 1978; Ruttner 1988) and this these are the species that would be present in the area at that time (and today). The beeswax lipid profile from this subspecies or the more Northern subspecies would be indistinguishable. However, we recognize that it is not possible to be entirely sure which sub species these beeswax lipids originate from so we have added the following comment (Lines 195-196) highlighted in bold below.

In Africa, there are ten subspecies of *Apis mellifera*⁴⁰, with *Apis mellifera adansonii* being regarded as the 'indigenous' Western Africa subspecies^{41,42}, **making it the likely candidate for the Nok beeswax lipids.**

p. 16, lines 293-294: « Finally, the invention of pottery represents a further stage of food processing for human groups, in this instance, allowing controlled heating of the comb to separate the wax and honey. » Ceramic vessels are not essential to separate beeswax from honey. For instance, the film of Eric Valli and Diane Summers on honey hunters of Nepal shows people who separate these two substances with their hands. This film could be mentioned and discussed in the paper for the practices without pottery.

Whilst we agree that heat is not necessary to separate the comb and honey, our point is that the controlled use of heat, made possible by ceramic technology, does facilitate the process and may represent a further stage in food processing by humans. Nonetheless, we have amended this sentence as follows (line 311):

Finally, **although honey can be squeezed from the bee combs by hand or sieved through a mesh**, the invention of pottery represents a further stage of food processing for human groups, in this instance, allowing controlled heating of the comb to separate the wax and honey.

There are several films available on honey hunting in Nepal and other places, including the well-known film by Eric Valli and Diane Summers, but we prefer not to discuss these in the

paper. We would note that very often these films do show the separation of combs and honey through heating.

And a last thing: in the abstract as well as on pages 7 and 8, the authors indicate that they analysed 458 potsherds but page 6 talks about 450 potsherds. I guess that this is a mistake and that 450 has to be replaced by 458.

We had stated 'over 450 potsherds' on page 6 but agree that it would be better to state '458 potsherds' as a matter of consistency so this has been changed.

Reviewer #3 (Remarks to the Author):

This study applies a well-established methodology to pottery sherds belonging to the Nok culture of West Africa. A total of 458 potsherds from 12 sites were investigated. Lipids were only recovered from < 20% of the sherds. Of this group, 25 lipid profiles are consistent with beeswax, although the quality and abundance of the biomarker distributions is variable. These results are discussed in the context of beeswax and honey use.

Beeswax has been identified in archaeological contexts using GC-MS on numerous occasions over the past 25 years. The methodology is routine as is the chemical identification of beeswax. Mixtures of beeswax with other lipids is explored. Much of the discussion draws on documentary evidence for the diverse uses of honey and beeswax in West Africa. The analytical data presented here simply provides a range of possibilities in terms of wider use and roles of beeswax and honey among these communities. **There is no information as to whether these roles changed over the 1500 year period of the Nok culture.**

We have added in the following section (lines 218-228) to provide information on the level of beeswax processing across the Nok chronological span:

The 25 potsherds containing compounds suggestive of the presence of beeswax/honey come from three periods: Early Nok, Early Middle and Later Middle Nok, perhaps not surprisingly, as these categories comprised the largest numbers of sherds sampled and lipid-containing sherds. These data demonstrate that beeswax and/or honey exploitation occurs throughout the Nok culture, but, interestingly, no biomarkers for beeswax were found in Common Era sherds. Indeed, only 8% ($n=5$, Table 1) of these sherds yielded lipids suggesting either that, overall, Common Era pottery was not used as much or lipid preservation conditions at that time were less favourable. Overall, beeswax processing comprises 38% across all periods but is the most frequent commodity processed in Early Nok pottery (55%, $n=11$, Table 2), decreasing somewhat in Early and Later Middle Nok pottery (38% and 38%, $n=9$ and $n=5$, respectively, Table 2).

I am happy to recommend publication in Nature Communications, subject to the following points that the authors should consider:

(i) Beeswax/honey. To avoid confusion, the authors should distinguish more carefully the identification of beeswax, reliably presented in the manuscript, and the presumed presence of honey, which due to its sugar content will rarely survive in archaeological contexts. There is an example on page 10/line 171 (These results unambiguously confirm these vessels contained beeswax/honey residues). Simply put, the results confirm the former but not the latter. There are many examples from other contexts where beeswax was used as a product in its own right. There is a comment in lines 251-252 but greater clarity is needed overall.

This was also noted by reviewer 2 and we have made amendments to read beeswax processing throughout the manuscript as requested. We have already provided discussion in the text of whether the beeswax lipids relate specifically to its use for technological purposes or whether it acts as a proxy for honey use and consumption.

(ii) Chronology. The Nok culture spans 1500 years. Table 1 breaks down this down into five consecutive periods. For completion, can the authors include the dates of each of the chronological periods?

Table 1 has been changed to show chronological periods. We have also added in discussion of beeswax recovery across the chronological span of the Nok culture (see above, lines 218-228).

(iii) Pottery vessels. The authors do not include any information of the form/type of the vessels, especially if there is a relationship between form and residue type. Can the authors provide more information here? Could some of the pottery vessels have served as beehives?

The Nok pottery assemblage is quite limited and all vessels sampled comprised everted rim vessels. Thus, no relationship between form and residue type could be established. We have added this to the text (lines 117-118).

Vessels sampled comprised everted rim pots with body diameters of c. 20 – 30 cm, the common form found within the Nok pottery assemblage.

It is a very valid point that the vessels may have been used as beehives so we have added the following text (lines 303-310):

A further possibility is that the pots themselves may have been used as beehives (Fig. 6a and b), implying management of the bees. This is commonplace today in East and South Africa and also evidenced in modern-day Nigerian traditional beekeeping. Here, pottery hives are either placed on the ground or in trees, while other types of hives made from clay, mud, straw or bark are always placed in trees⁶² (Fig. 6c). Clay hives were also recorded in Nigeria, Burkina Faso, Malawi and Ethiopia^{62,65,69} and clay vessels with holes in them were used as beehives in Mozambique until the 1970s and are still used in Kenya today^{69,70}. However, Nok vessels are generally only 20-30 cm diameter, and likely too small to have been used for these purposes.

(iv) Uses of wax and honey. The authors should include information, pertinent to the West African context, on the medicinal uses of honey and wax. The authors don't mention propolis, which might also be included here. The text is rather dominated by relationships to food and drink.

Agreed. We have added the following text (lines 237-242).

Bee products, including honey, propolis, royal jelly and venom, comprising various bioactive properties, have a history of use for various medicinal purposes, both in West Africa and globally. For example, propolis has both antiseptic and anaesthetic properties and is often used as an ingredient in medicines, toothpastes, oral sprays and chewing gums and royal jelly is valued as a medicine, tonic or aphrodisiac, likely because it contains many insect growth hormones¹³.

Reviewer #4 (Remarks to the Author):

This manuscript by Dunne et al. uses well-developed chromatographic methods to characterize lipids absorbed in early Iron age potsherds of the Nok culture in the humid tropics of West Africa. A total of 458 sherds was analyzed, 66 preserved lipids, beeswax was securely identified in five sherds, probable in one, and possible in 19. The identification of beeswax in pottery is not novel, in general. The main point of significance identified in the abstract is that it the first identification in West African archaeological ceramics.

This manuscript is remarkably well-constructed and would require only minor revisions for publication as a report of interest to an archaeological community. Points that could be elaborated or revised that would make this report of wider interest are discussed in the remainder of this review.

1. Honey, bee larvae and adult bees are valuable source of energy and protein exploited by many African vertebrate species, including honey badgers (*Mellivora capensis*), chimpanzees (*Pan troglodytes*, McLennan, *Am J Phys Anth* 2015; Boesch et al. *J Human Evol* 2009), humans, and avian bee-eaters (*Meropidae*). Honey guide birds (*Indicator* sp.) may digest beeswax (Downs et al., 2002. *Comparative Biochemistry and Physiology A* 133, 125-134.), and these birds actively recruit humans and honey badgers to natural beehives (Wood et al. 2014, *Evolution and Human Behavior* 35, 540). The diversity of taxa that intensively exploit African honey bees, and commensal/symbiotic relationships among species, suggests a substantial time depth of human exploitation of honey bees, honey and beeswax- beginning long before pottery and perhaps before the australopithecines. Control of fire to generate smoke to repel bees while their hives are being plundered, may have been an important inflection point in the history of human honey consumption, and also processing of wax and honey wine fermentation.

Done. Discussed below – see below for reviewer summary.

2. Pottery vessels are not a prerequisite for honey fermentation. The Okiek hunter-

gatherers of highland Kenya often used leather bags for fermentation of honey, according to Blackburn.

Added in - line 268

3. Evidence for beeswax dating over 40 ka (thousand years) in Southern Africa (cited in the ms under review) and in southern Europe at approximately the same age (Sano et al. 2019. *Nature Ecology and Evolution* 3, 1409), suggests an important additional role for beeswax as a waterproofing component of gums and mastics for hafting stone tools. This is notable because most tree gums are water soluble, so glued composite artifacts could disassemble in the wet season.

As the reviewer states we have already mentioned the Southern African beeswax in the paper in the context of it being used to haft bone points. We would prefer not to mention the Sano et al 2019 paper as it relates to southern Europe and is not relevant in the context of African archaeology. However, we do already discuss the technological uses of beeswax in the paper. See lines 243-250 as follows:

Beeswax itself has been used for technological purposes since the Palaeolithic, with the earliest known use being as an **adhesive** at Border Cave, South Africa, *c.* 40,000 years ago, where a lump of carefully curated organic material, comprising a mixture of beeswax and *Euphorbia tirucalli* resin wrapped in vegetal fibers, was likely used to **haft a bone point**⁵⁷. Beeswax has also variously been used from prehistoric times as a **sealant or waterproofing agent** on Early Neolithic collared flasks in northern Europe⁵⁸, as a **lamp illuminant** in Minoan Crete³⁷ and **mixed with tallow, possibly for making candles**, in medieval vessels at West Cotton, Northamptonshire⁴⁶.

4. Human biology may also reflect a deep time relationship of humans and honey wine/beer. The alcohol and aldehyde dehydrogenase alleles predominant among Africans may reflect a substantial time depth for metabolism of alcohol, including honey-based fermented beverages. In contrast, Asian populations include many individuals with alleles of both genes that make ethanol consumption uncomfortable at best.

We prefer not to mention this. Should we have been arguing that the presence of beeswax/honey means that the vessels were solely used for the processing of alcoholic drinks then it would be relevant and useful to mention these studies but we cannot know whether the beeswax/honey in the vessels was used for technological purposes, medicinal reasons, for storing or processing honey as a foodstuff or making drinks, alcoholic or otherwise, so have presented the various possibilities and evidence for the various options.

5. The mode of acquisition of honey is referred to in the paper title, abstract and main text as "honey-hunting", and is supported by references (5-13) to collecting honey (not hunting honey) from natural hives, mainly by hunter-gatherers. Natural hives are exploited universally by African honey collectors. However, many African societies manufacture hives and are prolific beekeepers and 'cultivators', notably the Okiek

hunter-gatherers of Kenya discussed in ref. 9. Traditional manufactured beehives dangle from trees almost everywhere in Sub-Saharan Africa. Ethiopian peoples have well-developed honey-based alcoholic beverage production and manufactured hives. References to beekeeping rather than honey 'hunting' are not cited until ref 62; Ref 64 refers to bee husbandry in Nigeria, and ref 65 to beekeeping in western and northern Africa. Therefore if African foragers and agriculturalists are prolific bee-keepers and honey collectors, then "hunting" is a misnomer. It diminishes the scale and technological and social complexity of well-developed African hunter-gatherer and agriculturalists honey production and exploitation systems. Moreover, it implies that the agricultural Nok peoples were not beekeepers.

Agreed - see below for reviewers summary.

6. Most interesting for the paper under review, is Irvine's review of traditional African beekeeping (1957, *Bee World* 38, 117), which describes pottery beehives that are smeared with beeswax inside to attract bee colonies. This practice could be considered one form of beeswax processing identifiable with lipid analyses. Beeswax traces in Nok sherds could also indicate pottery hives. One example of a ceramic vessel interior with traces of where the brood/honey combs were once attached is shown in fig 10 of ref 9 by Russell and Lander 2015. This kind of trace is clearly evidence for beekeeping rather than honey processing.

This is a useful point and we have added in discussion on the Nok pots possibly being used as beehives (lines 303-310). We have also added a figure showing the image of a possible pottery beehive (from the Russell and Lander 2015 paper, supplied by Gavin Whitelaw). However, it should be noted that ceramic vessels left on the ground may not necessarily have been used deliberately but may have been coincidentally colonised by bees, as is thought to be the case with the vessel in the Russell and Lander 2015 paper.

7. The local archaeological context could be described briefly, particularly the first appearance of pottery in this region. Mali is mentioned for the earliest pottery in Africa. Ref. 29, which I cannot access, apparently discusses evidence for millet agriculture at 3500 bp, described as contemporary with the Nok Culture. Kintampo Neolithic sites are relevant predecessors in this region. Some Kintampo sites have abundant evidence for exploitation of lipid-rich *Canarium* tree and palm nuts in the form of charred botanical remains. Lipid analysis of Kintampo ceramics might provide evidence for oil extraction or other nut processing practices, as well as earlier evidence of honey and beeswax processing in the West African savanna woodlands.

The local archaeological context has already been described on lines 76-85 and 90-98. We would prefer not to mention the Kintampo culture as we would argue that Kintampo Neolithic sites are not relevant predecessors in the region. Kintampo is earlier than Nok but there is absolutely no evidence for any connection between the two complexes or even Kintampo being a predecessor of Nok. Ghana and Nigeria are quite some distance from each other and the Kintampo culture is much further west. By the time Nok emerges, Kintampo is almost over.

We would note that lipid analysis of Kintampo pottery would be an entirely separate project!

However, we have added to our text as follows (line 280):

The first farmers appear in this region (**likely from the North**) at around 1500 BC.

We have already mentioned the adoption of exploitation of canarium on line 282:

‘gained their knowledge of bee behaviours through cultural contacts with indigenous hunter-gatherers, **similarly to the adoption of local *Canarium schweinfurthii* fruits, a valuable source of fat²⁹**’.

8. Figure 1 shows color variations that might represent elevation. Please explain in the caption. what is being shown with this color scheme.

Elevation has been added to the figure.

9. Figure 2 shows three classic Nok effigy vessels. While these figures are impressive, it might be more relevant to show an image of a traditional hive, particularly a pottery hive. These Nok effigy ceramics would, however, be a good journal cover photo.

We have added in a figure (Fig. 6) showing two ceramic hives and a traditional Nigerian straw hive.

10. More discussion regarding how beeswax become impregnated in porous ceramics would be welcome. Many questions arise:

Does storage of strained honey without wax combs deposit wax lipids in the vessel walls?

If wax and other hydrophobic molecules have lower densities than honey, does absorption occur preferentially at the level of the liquid surface, at the vessel neck (as Evershed and others have shown for animal fats)?

If honey is brewed into mead then does the low heat used to accelerate fermentation increase lipid absorption at the vessel neck?

Does ethanol act as a solvent for some of the lipids, facilitating absorption or volatilization?

If combs were systematically melted at higher temperatures to purify the wax, does this alter the proportions of lipids of different molecular weights and volatilities?

Does wax processing concentrate lipids in the vessel bottom rather than at the neck?

Does honey have non-wax lipids or is it pure carbohydrate and water?

Do pollen and insect cuticle lipids have distinctive chromatographic signatures?

These question beg for answers through systematic experiments and analysis of ethnographically documented ceramics used for honey processing.

Please see below for the reviewer's summary questions and our answer to this.

10. The chromatographic evidence for beeswax is not evidence for honey itself. Honey could be eaten raw from the comb, or pressed, and honey-free beeswax could then processed by melting for sealing, waterproofing, and as an additive to gums and mastics for waterproofing, among other uses.

This comment was also made by reviewers 2 and 3. We have changed the text to reflect that it is beeswax that is being processed in the vessels but we also refer the reviewer to our comments in the article where (lines 243-250, see below) we do indeed discuss the use of beeswax for technological processes.

Beeswax itself has been used for technological purposes since the Palaeolithic, with the earliest known use being as an adhesive at Border Cave, South Africa, c. 40,000 years ago, where a lump of carefully curated organic material, comprising a mixture of beeswax and *Euphorbia tirucalli* resin wrapped in vegetal fibers, was likely used to haft a bone point⁵⁷. Beeswax has also variously been used from prehistoric times as a sealant or waterproofing agent on Early Neolithic collared flasks in northern Europe⁵⁸, as a lamp illuminant in Minoan Crete³⁷ and mixed with tallow, possibly for making candles, in medieval vessels at West Cotton, Northamptonshire⁴⁶.

The chromatograms in figures 4 and 5 could be more readily interpretable for this reviewer if chromatograms of beeswax reference samples were also shown, perhaps in a supplemental information section.

Reference chromatograms for modern beeswax have been shown in full in Regert et al 2001 and are fully referenced in the text.

In summary, this paper could be improved for the general readership of Nature journals by: (1) discussing more of the evidence suggesting a considerable time depth for honey consumption in Africa, (2) discuss beekeeping practices as wells as honey collecting (rather than hunting), (3) consider a broader suite of documented practices that could leave traces of beeswax in potsherds, and (4) describe how lipid types, proportions and distributions vary within vessels with different processing practices.

Regarding point (1) made by the reviewer **discussing more of the evidence suggesting a considerable time depth for honey consumption in Africa** we thank him for this suggestion and have added the following text to our introductory paragraph (lines 46-52). We agree that discussion of the long relationship between humans and honey exploitation has broadened and strengthened the paper.

Honey, a rare source of sweetness, was likely a much sought-after foodstuff for much of human history. **Recognition that bee products, including honey and larvae, offered a high-quality source of dietary energy, fat and protein explains the long history of bee**

exploitation in the hominin lineage^{1,2}. Honey is energetically dense^{3,4} and easy to consume and digest and thus may have contributed to potential links between nutrition and neural expansion of the enlarging hominin brain³. Our closest living relatives, the chimpanzees (*Pan troglodytes*), forage for honey (and brood) when it is available, as do baboons and other great apes⁵, suggesting the importance of honey extraction in the emergence of complex tool use in hominoids⁶.

Human biology may also reflect a deep time relationship of humans and honey wine/beer. The alcohol and aldehyde dehydrogenase alleles predominant among Africans may reflect a substantial time depth for metabolism of alcohol, including honey-based fermented beverages. In contrast, Asian populations include many individuals with alleles of both genes that make ethanol consumption uncomfortable at best.

As noted, we prefer not to mention this. Should we have been solely arguing for the processing of alcoholic drinks then it would be useful to mention this but we cannot know whether the vessels were used for food, making drinks etc so have presented the various possibilities and evidence for them.

With regard to point (2) raised by the reviewer ‘**discuss beekeeping practices as well as honey collecting (rather than hunting)**’

We have amended the paper to read honey collecting rather than honey hunting and have now made reference to the pots themselves possibly being used as beehives (although we think that unlikely as they are too small).

However, we have now mentioned (on lines 303-310) the possibility that the bees were managed:

A further possibility is that the pots themselves may have been used as beehives, implying management of the bees. This is commonplace today in East and South Africa and also evidenced in modern-day Nigerian traditional beekeeping. Here, pottery hives are either placed on the ground or in trees, while other types of hives made from clay, mud, straw or bark are always placed in trees⁶² (Fig. 6c). Clay hives were also recorded in Nigeria, Burkina Faso and Ethiopia⁶⁵ and clay vessels with holes in them were used as beehives in Mozambique⁶⁹ until the 1970s and are still used in Kenya today⁷⁰. However, Nok vessels are generally only 20-30 cm diameter, and likely too small to have been used for these purposes.

We have also added this text to our conclusion (line 315):

Given the early use of pottery in prehistoric West Africa, this first identification of beeswax/honey residues, **whether through honey hunting or beekeeping**, in an early farming context 3500 years ago hints at a much older history of exploitation.

With regard to point (3) raised by the reviewer ‘**consider a broader suite of documented practices that could leave traces of beeswax in pots/herds**’

We consider that we have discussed every possible option regarding the suite of documented practises that could deposit traces of beeswax in pots/herds including its use for technological

purposes, as a preservative for foods such as meat, for medicinal purposes, for storing or processing honey or making either alcoholic or non-alcoholic drinks and finally that the pots may themselves have been used as beehives.

With regard to point 4 raised by the reviewer **‘describe how lipid types, proportions and distributions vary within vessels with different processing practices’**. The reviewer also notes that the series of questions he poses above **‘These question beg for answers through systematic experiments and analysis of ethnographically documented ceramics used for honey processing’**.

We welcome Professor Ambrose’s comments on our paper but these points go beyond the remit of our study and, more importantly, the scope of questions that can realistically be answered by organic residue analysis. The principle basis of assessing the importance of beeswax in the pottery we have studied here is the numbers of vessels in the assemblage containing a beeswax lipid profile, which is higher than any other pottery assemblage we have studied in 30 years of research. Going beyond this would be moving to over-interpretations of the specifics of these residues, which we are reluctant to do. Our interpretations as they stand are perfectly commensurate with the data presented and the archaeological context.

Interestingly, we have addressed questions of the type raised by Professor Ambrose in relation to the very commonly occurring animal (and plant) based organic residues seen in archaeological pottery (see Charters *et al.* 1993; Charters *et al.* 1997; Evershed 2008; Reber *et al.* 2019; Miller *et al.* 2020). These papers form the basis of the interpretations of such residues and we have complemented these with ethnographic work where possible ((Evershed 2008; Dunne *et al.*, 2018). But, at this stage, there is nothing to be gained from going further with extensive experimental work or ethnographic studies, which will take years to complete and will add little to the interpretations given in our paper, especially given the limitations of what we can realistically discern from archaeological organic residues. Professor Ambrose should be reassured by the work on beeswax lipids and degradation experiments of modern beeswax carried out by Regert *et al.* 2001, the conclusions of which entirely concur with our ancient beeswax distributions. To the best of our belief, no systematic work on ethnographic beeswax lipids has been carried out (although this would be a fascinating research project!), however, the reference beeswax used in the papers of Evershed *et al.* 1997 and 2003 was sampled from a 150 year old ethnographic beehive from Crete. The distribution of compounds seen is entirely consistent with modern beeswax and the altered beeswax lipid profiles seen in the Nok pottery.

Reviewed by Stanley H. Ambrose

Reviewers' Comments:

Reviewer #4:

Remarks to the Author:

This manuscript by Dunne et al. has addressed most but not all issues and suggestions for revision by the reviewers. I will concentrate on the most significant ones that remain to be addressed.

The paper title begins with "Honey hunting..." Hunting is a behavioral inference that will be read as the most important conclusion of the study if it remains in the title. "Hunting" goes beyond the scientific evidence presented in this study. The authors state in their response to my first review that they prefer not to speculate about practices such as beverage fermentation and other behaviors. They state:

"...points go beyond the remit of our study and, more importantly, the scope of questions that can realistically be answered by organic residue analysis. ... Going beyond this would be moving to over-interpretations of the specifics of these residues, which we are reluctant to do. Our interpretations as they stand are perfectly commensurate with the data presented and the archaeological context."

Archaeologists set a high bar for standards of evidence for interpretation of hunting. For example, stone tool marks on bones from Pleistocene sites are no longer assumed to be evidence for hunting. Archaeologists generally acknowledge that the mode of death of the animals cannot be demonstrated conclusively from tool marks on bones. Rather, they frame this evidence in terms of butchery, and early versus late access to carcasses (relative to non-human predators and scavengers), and scavenging versus hunting. If the authors state that they prefer not to speculate about behaviors beyond the organic residue analysis, then they should apply this standard consistently to inferences such as hunting. Honey collecting and beekeeping are among the likely alternatives to hunting, but the residue analysis does not provide evidence to choose hunting over other alternatives. Honey hunting is therefore not an interpretation that is "... perfectly commensurate with the data presented and the archaeological context." See specific recommendations for lines 75-77 below.

Hunting is an inappropriate term in the broader sense because immobile resources such as plants are gathered or collected rather than hunted. Although both terms are widely used for exploitation of wild beehives consistency and accuracy in terminology is obviously important in science. The assumption of honey hunting in the archaeological context of an agricultural society where beekeeping may have been practiced is problematic. As noted in my first review, the evidence for beekeeping rather than hunting is cited only toward the latter half of the first submission of this manuscript. This separation of hunting from beekeeping in the manuscript remains unacceptable. Alternative hypotheses for mode of acquisition should all be discussed at the beginning of the manuscript rather than as an afterthought. The text should state whether they can infer any of these modes of acquisition from the biomolecules.

Considering the absence of evidence for mode of procurement of beeswax, and the unusually high frequency of beeswax esters in this pottery assemblage, a more appropriate title would be something like: Intensive use of beeswax (in the Neolithic era in) in West Africa (3500 years ago). One but not both of the parts in parentheses would provide the temporal context. If the sherds span several phases of the Nok culture, then a title with one date may be inaccurate.

Abstract line 31: If beeswax was "processed", then what kind of processing is "commensurate with the data presented"? Did processing involve heating, mixing with other organic compounds or fermentation? What kinds of differences in chemical composition between the residues extracted from Nok ceramics compared to modern unprocessed beeswax suggest processing? If processing cannot be differentiated from diagenesis then "processing". Processing is also mentioned on line 127, and the same question applies for these plant residue compositions.

Abstract line 33: "Honey collecting" is a more appropriate term than honey hunting. Please be

consistent in terminology. Honey collecting and "beeswax processing" are different behaviors. Characterizing the mode of procurement as collecting implicitly excludes the possibility of beekeeping. Procurement, acquisition, collection and use are among the more neutral terms.

Lines 52, 53, 57, 59, 62, 63 and 75 use the following seven terms for honey acquisition: forage, extraction, collected, exploitation, honey-gathering and honey-hunting and wild harvest.

Please choose one term consistently for procuring or acquiring honey from natural hives. Marlowe et al. 2014 use 'acquire' (and also 'collect') for human foragers and other apes, though their distinction is not clear.

Line 71: Please cite appropriate general review for ethnographic uses by foragers such as Marlowe et al. 2014, cite another for beekeeping, and another for honey-based drinks.

Lines 75-77: Nok represents an agricultural economy, so references to ethnographic descriptions of honey exploitation and beekeeping by settled food producers is appropriate. All of the references cited here are for acquisition from natural (wild) hives. Refs 60-63 refer to beekeeping rather than wild honey collecting and should be cited here.

Line 114: How many honey-producing bee species exist in West Africa? Hepburn and Radloff identify two *Apis mellifera* subspecies and a hybrid form in Nigeria. Identifying the *adansonii* subspecies seems to be based on geography rather than lipid analysis. Other reviewers have noted that this is an unwarranted inference. These are not the only honey-producing African bees. Marlowe et al. tabulate honey collection from seven bee species in Tanzania, two of which make nests underground. How many of these species are also found in West Africa? Does the wax of each species have a distinctive diagnostic lipid profile. Hepburn and Radloff (ref 40) review species and subspecies differences in lipid composition, as do Beverly et al. 1995.

Line 138: Are wax esters with WE/HWE >C40 more diagnostic of beeswax than the C23-35 n-alkanes, which are also common in C4 plants? Most (18) of the samples listed in Table 2 as possible rather than definite beeswax do not have WE and HWE >C40. Only 5 sherds are listed as unambiguous rather than possible beeswax, and another as "probable beeswax" due to trace amounts of heavy wax esters. They all have C>40. This raises questions about the reviewer response statement that "The principle basis of assessing the importance of beeswax in the pottery we have studied here is the numbers of vessels in the assemblage containing a beeswax lipid profile, which is higher than any other pottery assemblage we have studied in 30 years of research." It seems from the data presented, that five out of 66 potsherds (7.6%) with lipid residues are securely identified as beeswax. Of the total of 458 sherds analyzed, 1.1% have unambiguous beeswax residues. Are these percentages of sherds with C>40 the highest among assemblages studied over 30 years?

Identification of samples that lack the wax esters with C>40 as possible beeswax requires further justification. Regert et al. (2001) state that "volatility decreases with increasing molecular weight" (p 566), and on p. 561 discuss preferential loss of small n-alkanes rapidly at 100°C and over a few months at 60°C, and similar findings in other studies, including total loss (Heron et al. 1994). If the wax esters are most resistant to loss, then their absence while lower molecular weight compounds are recovered, seems inconsistent with identification as beeswax.

Lines 255-259: Honey can be stored raw for years. Are there ethnographic accounts of cooking honey? Processing is mentioned earlier in this ms. The chromatogram of heated wax (Regert et al. 2001, fig 7) shows even-numbered long-chain linear alcohols (C26-C34) are produced. If no unambiguous markers of thermal alteration are identified in the Nok pottery then processing vs storing are untested hypotheses. Processing is stated without qualification on lines 31, 110, 112 and 140. The question remains as to whether storage of honey could deposit wax lipids, and whether thermal identification can be identified.

Line 266: delete "ethnolinguistic".

Line 267: Marlowe et al. (ref 2) describe honey collection mainly but not exclusively from baobab trees. Honey of other bee species is collected from other tree species and also from underground nests of other bee species.

In conclusion, this review has focused on areas where terminology is used imprecisely and inaccurately, and multiple cases where inferences are made regarding the human behaviors associated with how wax residues were deposited in Nok ceramic vessels, and whether wax and/or honey were processed. Other reviewers noted that the loose use of beeswax/honey throughout the first draft was inappropriate because biochemical identification of honey itself is not presented. A conservative approach to inferring human behaviors from the analytical evidence of beeswax is exemplified by Heron et al. (1994). Interpretation of Nok pottery residues should follow their example if the authors do not want to go beyond "the scope of questions that can realistically be answered by organic residue analysis."

Stanley H. Ambrose

Response to reviewer #4

As we noted in our response to the editor on October 21st we were very appreciative of the four reviewer's comments and recommendations which we felt had helped strengthen the manuscript. We fully agree that our usage of terminology with regard to beeswax/honey processing was, on occasion, imprecise – a point raised by the majority of the reviewers. Although we believed we had addressed this in the first review, we are very grateful to Professor Ambrose for his further insights. We have thus made additional changes to the terminology (where appropriate) as requested and we'd like to thank Professor Ambrose for his valuable input. We hope that he will feel that the manuscript is much improved. We feel it will be of great interest to the community.

Below we address (in red text) the remarks of Professor Ambrose - Reviewer #4 (in normal text). Please note that quotations of text from the paper are shown in Times New Roman, to contrast with the Segoe UI text used by the reviewer and in our responses.

Reviewer #4

This manuscript by Dunne et al. has addressed most but not all issues and suggestions for revision by the reviewers. I will concentrate on the most significant ones that remain to be addressed.

The paper title begins with "Honey hunting..." Hunting is a behavioral inference that will be read as the most important conclusion of the study if it remains in the title. "Hunting" goes beyond the scientific evidence presented in this study. The authors state in their response to my first review that they prefer not to speculate about practices such as beverage fermentation and other behaviors. They state:

"...points go beyond the remit of our study and, more importantly, the scope of questions that can realistically be answered by organic residue analysis. ... Going beyond this would be moving to over-interpretations of the specifics of these residues, which we are reluctant to do. Our interpretations as they stand are perfectly commensurate with the data presented and the archaeological context."

Archaeologists set a high bar for standards of evidence for interpretation of hunting. For example, stone tool marks on bones from Pleistocene sites are no longer assumed to be evidence for hunting. Archaeologists generally acknowledge that the mode of death of the animals cannot be demonstrated conclusively from tool marks on bones. Rather, they frame this evidence in terms of butchery, and early versus late access to carcasses

(relative to non-human predators and scavengers), and scavenging versus hunting. If the authors state that they prefer not to speculate about behaviors beyond the organic residue analysis, then they should apply this standard consistently to inferences such as hunting. Honey collecting and beekeeping are among the likely alternatives to hunting, but the residue analysis does not provide evidence to choose hunting over other alternatives. Honey hunting is therefore not an interpretation that is "... perfectly commensurate with the data presented and the archaeological context." See specific recommendations for lines 75-77 below.

Hunting is an inappropriate term in the broader sense because immobile resources such as plants are gathered or collected rather than hunted. Although both terms are widely used for exploitation of wild beehives consistency and accuracy in terminology is obviously important in science. The assumption of honey hunting in the archaeological context of an agricultural society where beekeeping may have been practiced is problematic. As noted in my first review, the evidence for beekeeping rather than hunting is cited only toward the latter half of the first submission of this manuscript. This separation of hunting from beekeeping in the manuscript remains unacceptable. Alternative hypotheses for mode of acquisition should all be discussed at the beginning of the manuscript rather than as an afterthought. The text should state whether they can infer any of these modes of acquisition from the biomolecules.

Whilst we are happy to change the terminology from honey-hunting to honey-collecting in the manuscript, we would comment that we have used the term honey-hunting throughout this article as it is a commonly accepted term, used globally, to describe the practice of harvesting honey from wild bees. Indeed, Eva Crane, the world-renowned honeybee expert, herself called it 'honey hunting' in her seminal book "*Eva Crane: The world history of beekeeping and honey hunting*". Furthermore, Eva Crane published widely on bees in archaeological contexts and always used the term honey hunting.

Example publications (cited in the paper) showing honey-hunting is an accepted term are as follows:

Coppinger, C. R., Ellender, B. R., Stanley, D. A. and Osborne, J. Insights into the impacts of rural honey hunting in Zambia. *African Journal of Ecology* **0**(0), 1-5 (2019).

Pager, H. Rock Paintings in Southern Africa Showing Bees and Honey Hunting. *Bee World* **54**(2), 61-68 (1973).

Honey hunting is also the modern-day term used by the Food and Agriculture organisation of the United Nations – see report cited in the paper:

FAO *Non-Wood Forest Products, Bees and their Role in Forest Livelihoods*. (Rome, Food and Agriculture Organization, 2009).

From above - The assumption of honey hunting in the archaeological context of an agricultural society where beekeeping may have been practiced is problematic. As noted in my first review, the evidence for beekeeping rather than hunting is cited only toward the latter half of the first submission of this manuscript. This separation of hunting from beekeeping in the manuscript remains unacceptable.

We believe there may be a misunderstanding here in the reviewers interpretation of our intention in dealing with the issue of beekeeping or honey-hunting. The reviewer seems to be making the assumption that the beeswax residues we see originate from beekeeping – on the basis (we believe) that the Nok are thought to be agriculturalists.

However, we are clear that any inference that the Nok people were either beekeeping or honey hunting would be purely speculative and something that we cannot know (as we state archaeological evidence for beekeeping is extremely rare). Indeed, our colleagues in Frankfurt think it highly unlikely that Nok people would have practised beekeeping (this does not even happen in the area today, see below for information on modern day honeybee exploitation in the Nok region) and it is far more likely that they are either providing hives (see below) or opportunistically exploiting nests. But, as we cannot know which is the case we would prefer not to speculate, which is why we chose to present the different options. We believe this is the most parsimonious level of interpretation. We see the importance of the paper being the fact that the Nok people exploited beeswax, and, possibly, honey. How they did that is unknowable so we have discussed honeybee/beeswax exploitation by both foragers and farmers so as to cover both possibilities. It is for this reason that we thank the reviewer for suggesting the use of the term honey-collecting as opposed to honey-hunting as we agree it is far more appropriate for what the Nok people may be doing.

Interestingly, the following paragraph from the report from the Food and Agriculture Organization of the United Nations, Rome, 2009 (cited above) notes:

“Humans have devised many different ways to exploit bees for their honey and other products. Considering the wide range of bee practices still existing world wide and which can be categorized into three working definitions: honey hunting, beekeeping and a third category, named here as ‘bee maintaining’ which falls somewhere between honey hunting and beekeeping – where the beekeeper provides a nest site, or protects a colony of wild bees for subsequent plundering”.

As noted above, today, across much of Africa, and in particular, Nigeria, it is not bee-keeping that takes place in this area, as beekeeping practices are generally known or thought of in the Western world. Rather, artificial beehives made from log, bark, basket or clay are set in trees to attract honeybees (Ichikawa 1981; Terashima 1998; FAO 2009). In sub-Saharan Africa around 50-90% of hives placed in this manner are colonised by bees (Ruttner 1988, 200). Little, or no, management of the hives takes place, in contrast

to modern European beekeeping, mainly because of the tendency to abscond in tropical African honeybees. Once the colony has been in place for several months, harvesting of the combs takes place, sometimes destroying the nest. This usually takes place early in the rainy season, once the bees have exploited the freshly blossoming trees and before the heavy rains restrict the colony from collecting pollen (Seyfert 1930, 36-37; Mutsaers 1991, 6).

Traditionally, in the Nok region, locally available materials are used for the hives with local farmers building straw hives. Dried grass stems are woven onto a framework made from branches and the inside is lined with cow dung, often mixed with attractant herbs (Haruna pers. comm.). The hives are placed in trees in woodland patches within the more open surrounding landscape. The herbs help lure a swarm or an absconded colony of feral bees into the empty hive (Mutsaers 1993, Haruna pers. comm.).

To provide more information on modern-day honeybee exploitation practices, as a modern analogue, we have added into the text the following (lines 309-314):

“Little, or no, management of the hives takes place, in contrast to modern European beekeeping, mainly because of the tendency to abscond in tropical African honeybees. Once the colony has been in place for several months, harvesting of the combs takes place, sometimes destroying the nest. This usually takes place early in the rainy season, once the bees have exploited the freshly blossoming trees and before the heavy rains restrict the colony from collecting pollen^{63,66}”.

Respectfully, we feel the reviewer has perhaps misinterpreted our comments about **speculating about behaviours** ‘If the authors state that they prefer not to speculate about behaviors beyond the organic residue analysis, then they should apply this standard consistently to inferences such as hunting’.

We argue strongly that we cannot know for sure what activities the vessels were used for so we have provided, within the text, a full range of possible explanations for the beeswax within the vessels.

Considering the absence of evidence for mode of procurement of beeswax, and the unusually high frequency of beeswax esters in this pottery assemblage, a more appropriate title would be something like: Intensive use of beeswax (in the Neolithic era in) in West Africa (3500 years ago). One but not both of the parts in parentheses would provide the temporal context. If the sherds span several phases of the Nok culture, then a title with one date may be inaccurate.

We are clear in the paper that beeswax is present in the earliest Nok vessels, see lines 220-230 (repeated below). In fact, we have noted that beeswax is the most frequently found commodity (55%) in the Early Nok vessels (from 3500 years ago) so feel that it is perfectly appropriate to state ‘from 3500 years ago’.

“The 25 potsherds containing compounds suggestive of the presence of beeswax come from three periods: Early Nok, Early Middle and Later Middle Nok, perhaps not surprisingly, as these categories comprised the largest numbers of sherds sampled and lipid-containing sherds. These data demonstrate that beeswax, a direct indicator of bee exploitation, occurs throughout the Nok culture. Interestingly, no biomarkers for beeswax were found in (later) Common Era sherds. Indeed, only 8% ($n=5$, Table 1) of these sherds yielded lipids suggesting either that, overall, Common Era pottery was not used as much or lipid preservation conditions at that time were less favourable. Overall, beeswax occurred in 38% of lipid-yielding sherds across all periods but was the most frequent commodity processed in Early Nok pottery (55%, $n=11$, Table 2), decreasing somewhat in Early and Later Middle Nok pottery (38% and 38%, $n=9$ and $n=5$, respectively, Table 2)”.

Furthermore, our colleagues at Frankfurt, who excavated the Nok culture archaeological sites are clear that the term ‘Neolithic’ is not appropriate in this context.

Thus, we would be happy to change the title of the paper to:

“Honey-collecting in prehistoric West Africa from 3500 years ago”.

Abstract line 31: If beeswax was "processed", then what kind of processing is "commensurate with the data presented"? Did processing involve heating, mixing with other organic compounds or fermentation? What kinds of differences in chemical composition between the residues extracted from Nok ceramics compared to modern unprocessed beeswax suggest processing? If processing cannot be differentiated from diagenesis then "processing". Processing is also mentioned on line 127, and the same question applies for these plant residue compositions.

The word ‘processing’ where relating to beeswax is used 6 times in the manuscript and once in the abstract. We chose this word to cover generally how beeswax residues may come to be present in vessels, thereby encompassing a range of actions used for cooking foodstuffs in pots, from gentle heating through to prolonged boiling. It should be noted that we also used this term when discussing the identification of animal fats or plants in the Nok pottery. We regard it as a ‘catch-all’ and necessarily less specific term than ‘cooking’, which is quite specific. We intend the term to be broad in its meaning and one which encompasses a range of possible actions. However, the reviewer is quite correct to note that there are a couple of instances in the manuscript where the term processing should be substituted for a more appropriate term.

Where appropriate, the text has been changed:

Line 108 – changed ‘processing’ to ‘presence’ of beeswax in over one third

Line 137 - changed ‘processing’ to the ‘presence’ of beeswax

Line 254 and 256 – we had already written ‘arises as a consequence either of the processing (melting) of wax combs through gentle heating, leading to its absorption

within the vessel walls, or, alternatively, beeswax is assumed to act as a proxy for the processing (cooking) or storage of honey itself¹².

Abstract line 33: "Honey collecting" is a more appropriate term than honey hunting. Please be consistent in terminology. Honey collecting and "beeswax processing" are different behaviors. Characterizing the mode of procurement as collecting implicitly excludes the possibility of beekeeping. Procurement, acquisition, collection and use are among the more neutral terms.

Terminology, in terms of the use of the word 'collecting' has been addressed throughout the paper, where relevant (although see below).

Lines 52, 53, 57, 59, 62, 63 and 75 use the following seven terms for honey acquisition: **forage, extraction, collected, exploitation, honey-gathering and honey-hunting and wild harvest.**

The use of each of these seven terms is addressed in turn below.

1 and 2 Forage and extraction – used in the context of the following sentence:

“Our closest living relatives, the chimpanzees (*Pan troglodytes*), **forage** for honey (and brood) when it is available, as do baboons and other great apes⁵, suggesting the importance of honey **extraction** in the emergence of complex tool use in hominoids⁶.”

The first two terms queried here (according to the reviewer, lines 52, 53), namely **forage** and **extraction**, used in the context of chimpanzee honey consumption, are the terms used by the authors of the papers themselves and so are entirely appropriate. In the first paper (McLennan 2015) the author discusses chimpanzee **foraging** strategies i.e. “*seasonal foraging strategies of chimpanzees*”. In the second paper cited (Boesch *et al.*, 2009), the authors discuss **extraction** of honey by chimpanzees through the use of complex tool sets. Indeed, the word extract is used in the title of the second paper cited.

McLennan, M. R. Is honey a fallback food for wild chimpanzees or just a sweet treat? *American Journal of Physical Anthropology* **158**(4): 685-695 (2015).

Boesch, C., Head, J. and Robbins, M. M. Complex tool sets for honey **extraction** among chimpanzees in Loango National Park, Gabon. *Journal of Human Evolution* **56**(6): 560-569 (2009).

3 Collected

The word collected is now used throughout the text.

4 Exploitation

In the case of using the word exploitation we are using this in the context of 'exploitation of the honeybee' and feel this is perfectly appropriate terminology in this case.

5 honey-gathering - changed to honey-collecting

6 honey-hunting - changed to honey-collecting

7 wild harvest – we used this term for two reasons. We were discussing it in the context of modern-day honeybee exploitation “sources of income for local communities across much of Africa, through both beekeeping and **wild harvest**^{2,7,13}”.

Harvest or wild harvest is a term very commonly used today – for example, see the United Nations organisation publication we cite - FAO *Non-Wood Forest Products, Bees and their Role in Forest Livelihoods*. (Rome, Food and Agriculture Organization, 2009) and therefore we feel it is perfectly appropriate. Secondly, the phrase 'honey-collecting' is used in the sentence that follows “In the West African tropical rain forest, **collecting wild honey**, found in natural hollows in tree trunks and on the underside of thick branches, is a common subsistence activity¹⁵⁻¹⁷” so we would prefer not to substitute the phrase 'honey collecting' for 'wild harvest' in this instance as this phrase would not read well twice in such a close juxtaposition.

Please choose one term consistently for procuring or acquiring honey from natural hives. Marlowe et al. 2014 use 'acquire' (and also 'collect') for human foragers and other apes, though their distinction is not clear.

As above, we have changed terminology to honey-collecting throughout the paper to be consistent.

Line 71: Please cite appropriate general review for ethnographic uses by foragers such as Marlowe et al. 2014, cite another for beekeeping, and another for honey-based drinks.

Done, added Marlowe *et al*, 2014, Micheli 2013 and Platt 1955.

Lines 75-77: Nok represents an agricultural economy, so references to ethnographic descriptions of honey exploitation and beekeeping by settled food producers is appropriate. All of the references cited here are for acquisition from natural (wild) hives. Refs 60-63 refer to beekeeping rather than wild honey collecting and should be cited here.

Reference 15 cited here includes reference to both. Please see our comments above on beekeeping and honey-hunting.

Line 114: How many honey-producing bee species exist in West Africa? Hepburn and Radloff identify two *Apis mellifera* subspecies and a hybrid form in Nigeria. Identifying the *adansonii* subspecies seems to be based on geography rather than lipid analysis.

Other reviewers have noted that this is an unwarranted inference. These are not the only honey-producing African bees. Marlowe et al. tabulate honey collection from seven bee species in Tanzania, two of which make nests underground. How many of these species are also found in West Africa? Does the wax of each species have a distinctive diagnostic lipid profile. Hepburn and Radloff (ref 40) review species and subspecies differences in lipid composition, as do Beverly et al. 1995.

Reviewer 2 did indeed mention this and we addressed it in our previous response – see below:

With regard to our interpretation that the residues originate from *Apis mellifera adansonii* we have made this assumption because this sub-species is the ‘indigenous’ Western Africa subspecies (Fletcher 1978; Ruttner 1988) and thus these are the species that would be present in the area at that time (and today). The beeswax lipid profile from this subspecies or the more Northern subspecies would be indistinguishable. However, we recognize that it is not possible to be entirely sure which sub species these beeswax lipids originate from so we have added the following comment (Lines 195-196) highlighted in bold below.

“In Africa, there are ten subspecies of *Apis mellifera*⁴⁰, with *Apis mellifera adansonii* being regarded as the ‘indigenous’ Western Africa subspecies^{41,42}, **making it the likely candidate for the Nok beeswax lipids.**”

As honey from different bee species cannot be differentiated by lipid distributions (see text) then it is difficult to know how else we could attribute to species except by geography. As the reviewer is aware, Hepburn and Radloff are the definitive source for species of African honey bees. However, we are aware that hybridization likely occurs between *Apis mellifera adansonii* and *Apis mellifera jemenitica* so we have now added this to the text (line 199):

“Its distribution area of the wet tropical and equatorial zone along the coast of West Africa overlaps with that of *Apis mellifera jemenitica* subspecies to the north, who occupy the Sahel and the drier savannas of the North Sudan vegetation zone **and some hybridization between the two subspecies is thought to occur**⁴².”

We note that some exploitation of stingless bees does occur today (i.e. the other species mentioned by the reviewer) and have added this to the text (lines 203-204), although the lipid profiles suggest that it is products from *Apis* that are found in the pots.

“Stingless bees (*Meliponines*) are also exploited in Africa, although their honey yield is known to be much lower².”

With regard to the question does the wax of each species have a distinctive diagnostic lipid profile? We refer the reviewer to lines 199-202 where we clearly state (including the reference Beverly *et al.* 1995, which the reviewer cites):

“Significantly, it has been shown that there are no significant differences in the basic chemical composition of wax originating from different *A. mellifera* subspecies, only small variations related to the proportion of the predominant compounds^{45,46,47}, i.e. fatty acid esters (~67%), hydrocarbons (~14%), and free fatty acids (~13%).”

The references we cite here are as follows:

45. Tulloch, A. P.). Beeswax - composition and analysis. *Bee World* **61**(2), 47-62 (1980).

46. Beverly, M. B., Kay, P. T. and Voorhees, K. J. Principal component analysis of the pyrolysis-mass spectra from African, Africanized hybrid, and European beeswax. *Journal of analytical and applied pyrolysis* **34**(2), 251-263 (1995).

47 Frohlich, B., Riederer, M. and Tautz, J. Comb-wax discrimination by honeybees tested with the proboscis extension reflex. *Journal of experimental biology* **203**(10), 1581-1587 (2000).

A recent paper by Svečnjak *et al.* (2019) further confirms this:

“There are no significant differences in the basic chemical composition of wax originating from different *A. mellifera* subspecies, only small variations related to the proportion of the above mentioned predominant compounds (Beverly, Kay, & Voorhees, 1995; Fröhlich, Riederer, & Tautz, 2000; Tulloch, 1980)”.

Svečnjak, L., L. A. Chesson, A. Gallina, M. Maia, M. Martinello, F. Mutinelli, M. N. Muz, F. M. Nunes, F. Saucy and B. J. Tipple (2019). Standard methods for *Apis mellifera* beeswax research. *Journal of Apicultural Research* **58**(2): 1-108.

We had originally cited this paper but removed it to bring the references to the correct number (as the references we cite are the source literature and the Svečnjak *et al* 2019 paper is effectively a review). This reference could be re-instated if required.

We would point out that these authors all note that there are no significant differences in lipid composition – this, of course, refers to modern day wax. However, the lipid distributions we are looking at are typical of degraded waxes. We are careful to note this in the paper lines 179-184:

“Although it is relatively resistant to degradation, **the chromatographic profile of ancient beeswax often presents significant differences to that of contemporary beeswax**^{12,35}. For example, the free *n*-alkanols do not occur in fresh beeswax but are found in aged wax, due to hydrolysis of the wax esters. Furthermore, a preferential loss of shorter chain *n*-alkanes may induce a modification of the *n*-alkane profile through time³⁹”.

Line 138: Are wax esters with WE/HWE >C40 more diagnostic of beeswax than the C23-35 *n*-alkanes, which are also common in C4 plants? Most (18) of the samples listed in

Table 2 as possible rather than definite beeswax do not have WE and HWE >C40. Only 5 sherds are listed as unambiguous rather than possible beeswax, and another as "probable beeswax" due to trace amounts of heavy wax esters. They all have C>40. This raises questions about the reviewer response statement that "The principle basis of assessing the importance of beeswax in the pottery we have studied here is the numbers of vessels in the assemblage containing a beeswax lipid profile, which is higher than any other pottery assemblage we have studied in 30 years of research." It seems from the data presented, that five out of 66 potsherds (7.6%) with lipid residues are securely identified as beeswax. Of the total of 458 sherds analyzed, 1.1% have unambiguous beeswax residues. Are these percentages of sherds with C>40 the highest among assemblages studied over 30 years?

Whilst the *n*-alkanes are common constituents of plant lipids (see Dunne *et al.*, 2016), these are generally either seen in isolation or in combination with long-chain fatty acids in plant pottery lipids. In this instance, where the wax esters are not present (also see below) it is the highly diagnostic combination of *n*-alkanoic acids, *n*-alkanols and *n*-alkanes, in 'triplet' distributions (Fig. 4) which provide unambiguous evidence for beeswax, together with the presence of wax esters in the other potsherds. As we note below, a small number had low concentrations such that the wax esters likely did not survive. We argue that all sherds presented here provide evidence for beeswax, although not all yielded evidence of wax esters, due to their unique distributions, and can confirm that these are the highest (c. one third) among assemblages studied over 30 years. Together with the remarkable historic and ethnographic (and indeed modern-day) evidence for honey collecting and bee-keeping across Africa, we are confident that these pottery lipid residues provide secure evidence for early beeswax and honey exploitation in West Africa.

Identification of samples that lack the wax esters with C>40 as possible beeswax requires further justification. Regert *et al.* (2001) state that "volatility decreases with increasing molecular weight" (p 566), and on p. 561 discuss preferential loss of small *n*-alkanes rapidly at 100°C and over a few months at 60°C, and similar findings in other studies, including total loss (Heron *et al.* 1994). If the wax esters are most resistant to loss, then their absence while lower molecular weight compounds are recovered, seems inconsistent with identification as beeswax.

Whilst there are a number of samples that lack the wax esters, the lipid profiles of these samples, displaying the characteristic 'triplet' profile, comprise (lines 135-137):

"very distinctive series of even-numbered *n*-alkanoic acids (C₂₀ to C₃₂), *n*-alkanols (C₂₂ to C₃₄), and *n*-alkanes (C₂₃ to C₃₅). These are highly diagnostic and indicative of the presence of beeswax".

We note that reviewer 2 (a lipid residue specialist, and obviously very expert in beeswax residues) was clearly happy with the lipid discussion, making no comment on this.

As we note, concentrations are low in some of the samples, meaning that the wax esters may have undergone hydrolysis to their constituent parts (i.e *n*-alkanoic acids (C₂₀ to C₃₂), *n*-alkanols (C₂₂ to C₃₄), and *n*-alkanes) and may not have survived over archaeological timescales. See lines 155-156. These are still highly diagnostic.

“with the remainder likely being at too low concentration for preservation of the higher molecular weight compounds”.

Lines 255-259: Honey can be stored raw for years. Are there ethnographic accounts of cooking honey? Processing is mentioned earlier in this ms. The chromatogram of heated wax (Regert et al. 2001, fig 7) shows even-numbered long-chain linear alcohols (C₂₆-C₃₄) are produced. If no unambiguous markers of thermal alteration are identified in the Nok pottery then processing vs storing are untested hypotheses. Processing is stated without qualification on lines 31, 110, 112 and 140. The questions remains as to whether storage of honey could deposit wax lipids, and whether thermal identification can be identified.

There are many ethnographic accounts of heating honey, for example Platt (1955), Seyffert (1930) and Lewicki (1974), all cited in the paper.

Please also see above for our comments and subsequent amendments in relation to the term ‘processing’.

We believe we have discussed all the possibilities which could result in the deposition of beeswax within the vessels. For example (lines 253-259):

“The presence of beeswax in ancient pottery, identified through the complex lipid distributions discussed previously, most likely arises as a consequence either of the processing (melting) of wax combs through gentle heating, leading to its absorption within the vessel walls, or, alternatively, beeswax is assumed to act as a proxy for the processing (cooking) or storage of honey itself¹²”.

“The presence of high lipid concentrations in some Nok vessels suggests they may have been used in cooking or heating honey, possibly as an additive to other dishes, or storing it for consumption.”

We argue (above) that the high beeswax concentrations in some vessels suggest that honey may have been heated/cooked or beeswax heated/melted.

From reviewers comments above - The chromatogram of heated wax (Regert et al. 2001, fig 7) shows even-numbered long-chain linear alcohols (C₂₆-C₃₄) are produced. If no unambiguous markers of thermal alteration are identified in the Nok pottery then processing vs storing are untested hypotheses

We are not sure what the reviewer means here as we have discussed the clear presence of *n*-alkanols in several places – see lines 135-136:

“The remaining 25 lipid profiles (Table 2) comprised very distinctive series of even-numbered *n*-alkanoic acids (C₂₀ to C₃₂), *n*-alkanols (C₂₂ to C₃₄), and *n*-alkanes (C₂₃ to C₃₅), indicative of ...”

Lines 157-160

“Compounds dominating in these lipid profiles included the C₂₈, C₃₀ and C₃₂ *n*-alkanols and C₂₄ and C₂₆ *n*-alkanoic acids and, eluting at longer retention times, were a series of C₄₀–C₅₂ carbon number palmitic acid wax esters, maximising at C₄₆.”

Lines 163-166

“A further eight potsherds (NOK15, NOK93, NOK106, NOK127, NOK158, NOK246, NOK300 and NOK376, Table 2) also yielded lipid extracts containing the characteristic distributions of both *n*-alcohols (C₂₂-C₃₄ carbon number range) and *n*-alkanes (C₂₁ to C₃₁), described above.”

A final note of caution implicit in with our interpretations of the nature of “processing” is the inability to deconvolve changes in lipid distributions due to “processing” from those taphonomic/diagenetic alterations related to long term burial, which can manifest themselves in similar ways, e.g. hydrolysis of wax esters during burial producing palmitic acid and *n*-alkanols will be indistinguishable from hydrolysis occurring during vessel use. It is due to this equifinality that we have avoided overly specific interpretations regarding the precise mode of “processing”.

Line 266: delete "ethnolinguistic".

Deleted

Line 267: Marlowe et al. (ref 2) describe honey collection mainly but not exclusively from baobab trees. Honey of other bee species is collected from other tree species and also from underground nests of other bee species.

Agreed – although Baobab trees are the most commonly climbed in search of nests. Thus we have added in ‘mainly’ to this sentence.

“which they mainly collect whilst climbing Baobab trees²”

In conclusion, this review has focused on areas where terminology is used imprecisely and inaccurately, and multiple cases where inferences are made regarding the human behaviors associated with how wax residues were deposited in Nok ceramic vessels, and whether wax and/or honey were processed. Other reviewers noted that the loose use of beeswax/honey throughout the first draft was inappropriate because biochemical identification of honey itself is not presented. A conservative approach to inferring human behaviors from the analytical evidence of beeswax is exemplified by Heron et al. (1994). Interpretation of Nok pottery residues should follow their example if the authors do not want to go beyond "the scope of questions that can realistically be answered by organic residue analysis."

Stanley H. Ambrose

References

Ichikawa, M. (1981). Ecological and sociological importance of honey to the Mbuti Net Hunters, Easter Zaire.

Terashima, H. (1998). Honey and holidays: The interactions mediated by honey between Efe hunter-gatherers and Lese farmers in the Ituri forest.

Ruttner, F. (1988). Honeybees of Tropical Africa. In. *Biogeography and Taxonomy of Honeybees*. F. Ruttner. Berlin, Springer: 199-227.

Seyffert, C. (1930). *Biene und honig im volksleben der Afrikaner, mit besonderer berücksichtigung der bienenzucht, ihrer entstehung und verbreitung*. Leipzig, Voigtländer Verlag.

Mutsaers, M. (1991). Bees in their natural environment in southwestern Nigeria. *Nigerian field* **56**(1/2): 3-18.

Mutsaers, M. (1993). Honeybee husbandry in Nigeria: traditional and modern practices. *Nigerian Field* **58**: 2-18.

Reviewers' Comments:

Reviewer #2:

Remarks to the Author:

I think the authors answer to all the remarks of the reviewers and that the paper can now be published.